# Efficient Scheduling of Data Augmentation for Deep Reinforcement Learning

**Byungchan Ko**[*]
NALBI
kbc@nalbi.ai

**Jungseul Ok**
GSAI, POSTECH
jungseul@postech.ac.kr

## Abstract

In deep reinforcement learning (RL), data augmentation is widely considered as a tool to induce a set of useful priors about semantic consistency and to improve sample efficiency and generalization performance. However, even when the prior is useful for generalization, distilling it to RL agent often interferes with RL training and degenerates sample efficiency. Meanwhile, the agent is forgetful of the prior due to the non-stationary nature of RL. These observations suggest two extreme schedules of distillation: (i) over the entire training; or (ii) only at the end. Hence, we devise a stand-alone network distillation method to inject the consistency prior at any time (even after RL), and a simple yet efficient framework to automatically schedule the distillation. Specifically, the proposed framework first focuses on mastering train environments regardless of generalization by adaptively deciding which *or no* augmentation to be used for the training. After this, we add the distillation to extract the remaining benefits for generalization from all the augmentations, which requires no additional new samples. In our experiments, we demonstrate the utility of the proposed framework, in particular, that considers postponing the augmentation to the end of RL training. https://github.com/kbc-6723/es-da

## 1 Introduction

Deep reinforcement learning (RL) aims at finding a neural network to represent policy or value functions taking raw observation as input, of which the most common form in practice is visual data or images of high-dimensionality, e.g., video games [23], board games [29, 28], and robot controls [32, 18]. RL handling high-dimensional input often suffers from poor sample efficiency and generalization capability, mainly due to the curse of dimensionality [4, 15]. To overcome these issues, it has been widely considered to augment data based on prior knowledge that a set of transformations preserve the meaning or context of input observations, e.g., cropping out unimportant parts of images, and changing colors [20, 19, 21, 14]. On one hand, RL agent can be directly fed with the original and augmented data so that it implicitly learns a representation with the prior and improves the sample efficiency and generalization [20]. On the other hand, the prior knowledge in data augmentation can be explicitly distilled via a self-supervised learning, which introduces additional regularization to ensure consistency between responses to original and augmented inputs [25, 14].

However, data augmentation shows highly task-dependent effect in RL, and it is prone to generate severe interference with the training even when it truly conveys a useful prior to train and test environments [20, 25, 14]. We address the problem of alleviating the interference between data augmentation and RL training to improve *sample efficiency* for acquainting train tasks, and *generalization capability* for unseen test environments. This problem in (online) RL is more critical and challenging than that in supervised learning (SL) since the objective function and data distribution are time-varying in RL,

---

[*]This work was done while Byungchan Ko studied in GSAI, POSTECH.

36th Conference on Neural Information Processing Systems (NeurIPS 2022).

while they are not in SL. Indeed, according to [1, 10], it is sufficient to partly apply data augmentation just for a short period of SL at any time. Meanwhile, we empirically observe that the prior from data augmentation can be easily forgotten in RL of the non-stationary nature (see Section 5.3), i.e., the effect of augmentation is *time-sensitive*.

Based on our observation about the interference and the time-sensitivity, we propose two simple yet effective methods according to timings of data augmentation : **In**tra **D**istillation with **A**ugmented observations (InDA) and **Ex**tra **D**istillation with **A**ugmented observations (ExDA). Data augmentation beneficial for the sample efficiency needs to be applied over the entire RL training, i.e., InDA. Conversely, data augmentation useful only for the generalization should be postponed to the end of RL training, i.e., ExDA, so that we can minimize the interference in the training, while enjoying the benefit in the testing. InDA and ExDA are equipped with **D**istillation with **A**ugmented observation (DA). DA is a stand-alone self-supervised learning which enables us to induce the prior after RL training, and shows a relatively small interference with RL training by explicitly preserving the response of RL agent to the original input.

The best timing (InDA or ExDA) depends on traits of train task and augmentation. We hence suggest a framework of adaptive scheduling, named UCB-ExDA, that (i) first aims at maximizing the sample efficiency by adaptively selecting which or no augmentation to be used *during RL training*; and then, (ii) distill the priors from all the augmentations *after RL training*. Specifically, inspired by [25], we devise UCB-InDA for the adaptive selection in the first part by modifying the upper confidence bound (UCB) algorithm [3] for multi-armed bandit, where differently from [25], the set of arms includes all the augmentations and *the option to not augment*. In summary, UCB-ExDA is nothing but executing UCB-InDA followed by ExDA. Our experiment demonstrates the utility of the proposed framework.

Our contributions are summarized as follows:

(i) We devise DA (Section 4.1) which explicitly preserves the knowledge of RL agent so that enables distilling the consistency prior of augmentation into RL agent not only during but also after RL training, while other methods [20, 25, 14] need to be applied concurrently with RL training and thus show relatively strong interference in our experiments (Section 5.2).

(ii) We identify the simple yet effective timings of data augmentation: either InDA or ExDA (Section 4.2, Section 4.3), based upon the discovery of the time-sensitivity (Section 5.3) that has not been observed in SL [10], i.e., the proposed timings are effective particularly for RL.

(iii) We finally establish UCB-ExDA which automatically decides the best timing of augmentation, and demonstrate its superiority compared to others (Section 5.4). The advantage of UCB-ExDA is particularly substantial when the best strategy is ExDA postponing augmentation to the end of RL training.

## 2 Related Works

**Augmented experience in RL.** To solve the problem of poor generalization and sparse data, a popular approach is to fabricate virtual experiences and train the RL agent to learn with them. Domain randomization is a technique to produce such experiences from a simulator of a targeted system, [32, 24, 26]. Accurate simulators of practical systems are difficult to obtain, and it has limited applicability. However, visual augmentation has no such limit because the method uses simple image transformations such as cropping, tilting, and color jitter, although applications require a careful understanding of the targeted system to guide the design of an appropriate image transformer. A method of curriculum learning for domain randomization, in which the difficulty is gradually increasing [26] provided insights that coincide with some of our findings. However, we provide a further understanding of the types of visual augmentation that should be early or late during training.

Regularization from augmented data in vision-based RL has been implemented in various learning frameworks, including but not limited to representation [14, 31], self-supervised [25], and contrast [30]. One proposed algorithm [25] applies the UCB algorithm [3] to automatically select the most effective augmentations over RL training, where each augmentation is considered as an arm and then evaluate the effectiveness of augmentation by using a sliding window average. The idea of adapting augmentation concurs with our main message regarding the timing of augmentation. In [25], 'not augmenting' is not an option, whereas our findings indicate that it should be. In addition, [25] does not consider post augmentation followed by RL training, as in ExDA.

**Other time-sensitivity in deep learning.** During deep learning, the early stage of training often has a significant effect [7, 1]. Therefore, we devised time-sensitive methods that adapt to the progress of training, such as learning rate decay [35] and curriculum learning [33]. Golatkar *et al.* [10] studied such a time-sensitivity of regularization techniques for SL, where the effect of data augmentation in different time does not change much. We find that the time-sensitivity of augmentation can be significant in RL. This contrast may occur because of the non-stationary nature of RL, which SL does not have. Although a set of techniques originally developed for SL such as convolutional neural network, weight decay, batch normalization, dropout and self-supervised learning improve deep RL [16, 6, 22, 8, 30, 34, 13], a thorough study should be conducted before introducing a method from different learning frameworks, because we find the contrasting time-sensitivities of data augmentation. This spirit is also shared with an application [17] of implicit bias in SL [12, 2, 9] to RL.

## 3 Background

**Notation.** We consider a standard agent-environment interface of vision-based reinforcement learning in a discrete Markov decision process of state space $\mathcal{S}$, action space $\mathcal{A}$, and kernel $P = P(s_{t+1}, r_t | s_t, a_t)$ which determines the state transition and reward distribution. The goal of the RL agent is to find a policy that maximizes the expectation of cumulative reward $\sum_{t=0}^{t'-1} \gamma^t r_t$, where $t'$ is terminating time and $\gamma \in [0, 1]$ is discount factor. At each timestep $t$, the agent selects an action $a_t \in \mathcal{A}$ and receives a reward $r_t$ and an image $o_{t+1} = O(s_{t+1}) \in \mathbb{R}^{k \times k'}$ as an observation of the next state $s_{t+1}$. Data augmentation can be described by an image transformation function $\phi : \mathbb{R}^{k \times k'} \mapsto \mathbb{R}^{k \times k'}$ of which output is presumed to have the same or similar semantics of the input.

**Baseline RL algorithm.** Throughout this paper, we use Proximal Policy Optimization (PPO) [27] as a baseline, although we believe our findings and methods can be easily adjusted to others. PPO is a representative on-policy RL algorithm to learn policy $\pi_\theta(a \mid o)$ and value function $V_\theta$ of neural agent parameterized by $\theta$. Storing a set of recent transitions $\tau_t := (o_t, a_t, r_t, o_{t+1})$ in experience buffer $\mathcal{D}$, the agent is updated to minimize the following loss function:

$$L_{\text{PPO}}(\theta) = -L_\pi(\theta) + \alpha L_V(\theta) , \tag{1}$$

where $\alpha$ is a hyperparameter and canonical regularization terms are omitted. The clipped policy objective function $L_\pi$ and value loss function $L_V$ are defined as:

$$L_\pi(\theta) = \hat{\mathbb{E}}\Big[\min(\rho_t(\theta)\hat{A}_t, \text{clip}(\rho_t(\theta), 1 - \epsilon, 1 + \epsilon)\hat{A}_t)\Big] \tag{2}$$

$$L_V(\theta) = \hat{\mathbb{E}}\Big[\big(V_\theta(o_t) - V_t^{\text{targ}}\big)^2\Big] , \tag{3}$$

where the expectation $\hat{\mathbb{E}}$ is taken with respect to $\tau_t \sim \mathcal{D}$, we denote by $\theta_{\text{old}}$ the parameter before the update, $\rho_t(\theta)$ is the importance ratio $\frac{\pi_\theta(a_t|o_t)}{\pi_{\theta_{\text{old}}}(a_t|o_t)}$, and $\hat{A}_t$ is the generalized advantage estimation [27].

## 4 Method

In what follows, we present our methods: **D**istillation with **A**ugmented observations (DA; Section 4.1), **In**tra **DA** (InDA; Section 4.2), **Ex**tra **DA** (ExDA; Section 4.3), and then the adaptive scheduling frameworks based on UCB (UCB-InDA and UCB-ExDA; Section 4.4). DA is a stand-alone knowledge distillation method, which can be used at any time to instill the underlying prior of augmentation into a given RL agent. InDA and ExDA conduct either DA or PPO in each epoch but have different schedules (Figure 1), where InDA interleaves PPO and DA, whereas ExDA performs PPO first then DA. UCB-InDA adaptively decides which or no augmentation to be used in each DA epoch of InDA based on the UCB of estimated gain from each option. UCB-ExDA performs ExDA preceded by UCB-InDA.

### 4.1 Distillation with augmented observations (DA)

The underlying prior of augmentation can be infused by minimizing a measure of inconsistency between the agent's responses to original and augmented inputs (resp. $o_t$ and $\phi(o_t)$). For instance,

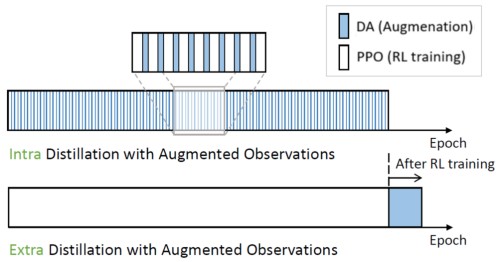

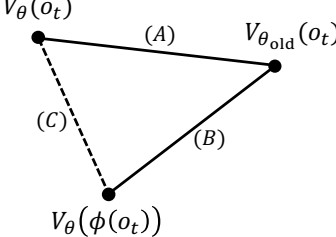

Figure 1: An illustration of InDA and ExDA    Figure 2: An illustration of distillation losses.

with PPO agent learning policy $\pi_\theta$ and value $V_\theta$, Raileanu *et al.* [25] proposes the following measure:

$$L_{\text{dis}}(\theta, \phi) := \hat{\mathbb{E}}_{o_t \sim \mathcal{D}} \left[ \text{KL}[\pi_\theta(\cdot|o_t), \pi_\theta(\cdot|\phi(o_t))] \right] + \hat{\mathbb{E}}_{o_t \sim \mathcal{D}} \left[ (V_\theta(o_t) - V_\theta(\phi(o_t)))^2 \right] , \quad (4)$$

which uses Kullback-Leibler divergence (first term) and mean squared deviation (second term) for policy and value inconsistencies, respectively. Noting that $L_{\text{dis}}(\theta; \phi)$ can be minimized to be zero by a constant response to all inputs, the distillation with Eq (4) can distort the RL agent, in particular, when applying it outside of RL training.

We hence devise a network distillation technique (DA) which *explicitly preserves* the RL agent's response to original input and thus is applicable even after RL training. DA distills the knowledge of RL agent $\theta_{\text{old}}$ into $\theta$ using not only original but also augmented observations. More formally, the loss of DA is written as:

$$L_{\text{DA}}(\theta, \phi; \theta_{\text{old}}) := L_{\text{dis}}(\theta, \mathbb{I}; \theta_{\text{old}}) + L_{\text{dis}}(\theta, \phi; \theta_{\text{old}}) . \quad (5)$$

We here denote the identity transformation by $\mathbb{I}$ such that $\mathbb{I}(o) = o$ for all $o$, and extend the definition of $L_{\text{dis}}$ in Eq (4) as follows:

$$L_{\text{dis}}(\theta, \phi; \theta_{\text{old}}) = \hat{\mathbb{E}}_{o_t \sim \mathcal{D}} \left[ \text{KL}[\pi_{\theta_{\text{old}}}(\cdot|o_t), \pi_\theta(\cdot|\phi(o_t))] \right] + \hat{\mathbb{E}}_{o_t \sim \mathcal{D}} \left[ (V_{\theta_{\text{old}}}(o_t) - V_\theta(\phi(o_t)))^2 \right] . \quad (6)$$

In Eq (5), the first term ensures that $\theta$ and $\theta_{\text{old}}$ behave identically for the original inputs, and the second one infuses the consistency prior. In Figure 2, $L_{\text{dis}}(\theta, \mathbb{I}; \theta_{\text{old}})$, $L_{\text{dis}}(\theta, \phi; \theta_{\text{old}})$, and $L_{\text{dis}}(\theta, \phi; \theta)$ graphically correspond to (A), (B), and (C), respectively. From this, it follows that minimizing $L_{\text{DA}}$ in Eq (5) eventually reduces $L_{\text{dis}}(\theta, \phi; \theta)$ in Eq (4). In addition, the minimization of $L_{\text{DA}}$ secures the responses of $\theta$ to the original inputs (which can be pre-computed to reduce computation cost) to the those of $\theta_{\text{old}}$, while the alternatives (e.g., (A)+(C): $L_{\text{dis}}(\theta, \mathbb{I}; \theta_{\text{old}}) + L_{\text{dis}}(\theta, \phi; \theta)$) does not and thus may generate interference with RL training. Our experiments (Section 5.2; Figure 4) justifies the design of DA by showing substantial advantage compared to the other alternatives such as DrAC [25] using $L_{\text{dis}}(\theta, \phi; \theta)$ in Eq (4). We note that this advantage becomes much clearer when a wrong augmentation is given, c.f., the supplementary material.

## 4.2    Intra distillation with augmented observations

InDA (Algorithm 1) alternates between minimizing $L_{\text{PPO}}$ and $L_{\text{DA}}$, i.e., PPO and DA are explicitly separated, whereas they are often executed simultaneously in other methods [25]. Such a clear separation reduces the interference [14]. We can control the frequency and timing of applying distillation with hyperparameters $I$, $S$ and $T$, where we perform DA after each $I$ rounds of RL training only if the current epoch $n$ is in the interval of $[S, T]$, i.e., $S$ and $T$ are the epochs to begin and terminate DA, respectively. We vary $S$ and $T$ to study the time-dependency of data augmentation. We denote InDA$[S, T]$ to indicate the period to apply DA, while we omit the indication when DA is applied over the entire period. We provide further details on InDA in the supplementary material.

| **Algorithm 1** InDA | **Algorithm 2** ExDA |
|---|---|
| **Require:** $N, I, \phi, S, T$ | **Require:** $N, M, \phi$ |
| 1: Initialize $\theta$ close to origin. | 1: Initialize $\theta$ close to origin. |
| 2: **for** $n = 1, 2, \ldots, N$ **do** | 2: //Pre-training phase with RL algorithm |
| 3:    // RL training | 3: **for** $n = 1, 2, \ldots, N$ **do** |
| 4:    Store sampled transitions to $\mathcal{D}$; | 4:    Store sampled transitions to $\mathcal{D}$; |
| 5:    Optimize RL objective $L_{\text{PPO}}(\theta)$ with $\mathcal{D}$; | 5:    Optimize RL objective $L_{\text{PPO}}(\theta)$ with $\mathcal{D}$; |
| 6:    // Distillation | 6: **end for** |
| 7:    **if** $n \in [S, T]$ and $\mod(n - 1, I) = 0$ **then** | 7: |
| 8:      Store $\theta_{\text{old}} \leftarrow \theta$; | 8: Store $\theta_{\text{old}} \leftarrow \theta$; |
| 9:      Minimize $L_{\text{DA}}(\theta)$ for $\mathcal{D}$, $\theta_{\text{old}}$ and $\phi$; | 9: // Distillation at the end of RL training |
| 10:    **end if** | 10: **for** $m = 1, 2, \ldots, M$ **do** |
| 11: **end for** | 11:    Minimize $L_{\text{DA}}(\theta)$ for $\mathcal{D}$, $\theta_{\text{old}}$ and $\phi$; |
| | 12: **end for** |

## 4.3 Extra distillation with augmented observations

ExDA (Algorithm 2) performs the distillation after the end of RL training, where the lengths of DA and RL training are parameterized by $M$ and $N$, respectively. We note that computational cost can be reduced by removing the value inconsistency measure $\hat{\mathbb{E}}_{o_t \sim \mathcal{D}} \left[ (V_{\theta_{\text{old}}}(o_t) - V_\theta(\phi(o_t)))^2 \right]$ from $L_{\text{dis}}$ in Eq (6) because the value consistency is not necessary for RL training nor distillation in the actor-critic framework and including it has a potential risk of generating additional interference. In the supplementary material, it is empirically verified that this reduction does not degrade RL performance. We leave more interesting details in the supplementary. For instance, one can consider re-initializing $\theta$ before starting DA as a part of exploiting the implicit bias [12, 17] to improve generalization. However, test performance is often dropped. This is mainly because the dataset to distill $\pi_{\theta_{old}}$ has much less diversity than that observed along the trajectory. Thus, we use no re-initialization for the experiments in the main paper.

## 4.4 UCB-based adaptive scheduling frameworks

The training benefit by augmentation differs depending on the task. This dependency complicates the choice of whether to use InDA or ExDA for augmentation. Hence, we devise an auto-augmentation method, called UCB-InDA, inspired by UCB-DrAC [25], where each augmentation is corresponded to an arm in multi-armed bandit (MAB) problem and assessed its potential gain in training with upper confidence bound (UCB) [3]. More formally, in UCB-InDA, the set of arms is the set of augmentations, $\Phi = \{\phi_1, \ldots, \phi_K\}$, which must include the identity function $\mathbb{I}$, i.e., the option not to augment. The inclusion of identity function is small but makes crucial difference than UCB-DrAC [25] since we observe that using augmentation sometimes needs to be postponed after RL training for the sake of better sampling complexity and test performance.

A round of MAB corresponds to every $I$ epoch of InDA, where we let $\phi_{k(s)} \in \Phi$ be the augmentation selected at the $s$-th round of DA. Let $G(s)$ be the average return, the sum of estimated advantage $\hat{A}$ and predicted value $V_\theta$, over $(I-1)$ epochs of PPO followed by the $s$-th DA. UCB algorithm assumes that each arm generates reward independently drawn from a fixed distribution, and estimates the empirical mean over the entire sampling process. However, in RL, the return $G(s)$ is non-stationary, so UCB-InDA computes moving average gain $\bar{G}_k(s)$, instead, taken over a certain number (chosen to be 3 in our experiment) of most recent rounds selecting $\phi_k$ as Raileanu *et al.* [25] proposed. Then, inspired by UCB1 algorithm [3], UCB-InDA selects action $k(s)$ at round $s$ as follows:

$$k(s) = \underset{k \in \{1, \ldots, K\}}{\arg\max} \ \bar{G}_k(s) + c \sqrt{\frac{\log(s)}{N_k(s)}} \qquad (7)$$

where $c$ is the UCB exploration coefficient, and $N_k(s)$ is the number of times selecting $\phi_k$ up to round $s$. We refer to the supplementary material for the hyperparameter choice. We remark that compared to UCB-DrAC [25], the proposed UCB-InDA has subtle but important differences, summarized in two folds: (i) the inclusion of identity transformation (i.e., no augmentation) and (ii) the distillation with

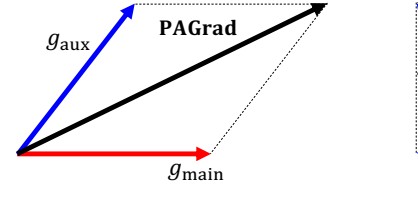 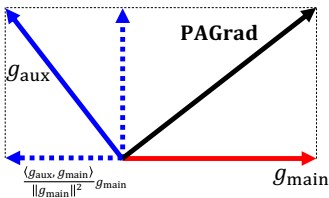

(a) non-conflicting: $\langle g_{\text{aux}}, g_{\text{main}} \rangle \geq 0$       (b) conflicting: $\langle g_{\text{aux}}, g_{\text{main}} \rangle < 0$

Figure 3: *An illustration of gradient conflicting and PAGrad.* We here let $g_{\text{main}}$ and $g_{\text{aux}}$ denote the gradients of main (shown in red; e.g., $\nabla L_{\text{PPO}}$) and auxiliary (shown in blue; e.g., $\nabla L_{\text{dis}}$) losses, respectively. In the computation of PAGrad (shown in black), the component of $g_{\text{aux}}$ conflicting to $g_{\text{main}}$ only when it exists (i.e., $\langle g_{\text{aux}}, g_{\text{main}} \rangle < 0$).

augmentation via InDA. The gain of each component is numerically studied in Section 5. Finally, we note that UCB-ExDA is nothing but UCB-InDA followed by ExDA.

### 4.5 PAGrad

Inspired by [36], we devise an alternative approach to reduce the interference by interpreting RL training with data augmentation as a multi-task learning, where $L_{\text{PPO}}$ and $L_{\text{dis}}$ correspond to the main and auxiliary task losses, respectively. In [36], the degree of conflict from task A to task B is estimated by the inner product of the gradient of task A and the negative gradient of task B, c.f., Figure 3. From this, we propose PAGrad (Projecting Auxiliary Gradient) to compute a modified gradient of $L_{\text{PPO}}$ excluding the conflict from the auxiliary gradient $\nabla L_{\text{dis}}$ to the main one $\nabla L_{\text{PPO}}$. Formally, PAGrad computes the gradient given as follows:

$$\nabla L_{\text{PPO}} + \left( \nabla L_{\text{dis}} - \frac{\min\{0, \langle \nabla L_{\text{dis}}, \nabla L_{\text{PPO}} \rangle\}}{\|\nabla L_{\text{PPO}}\|^2} \nabla L_{\text{PPO}} \right) , \tag{8}$$

where $\frac{\min\{0, \langle \nabla L_{\text{dis}}, \nabla L_{\text{PPO}} \rangle\}}{\|\nabla L_{\text{PPO}}\|^2} \nabla L_{\text{PPO}}$ is the components of $\nabla L_{\text{dis}}$ opposite to $\nabla L_{\text{PPO}}$ which may disturb optimizing the main objective $L_{\text{PPO}}$. Based on this, we devise DrAC+PAGrad that updates the model parameter toward the negative direction of (8). This is an alternative of InDA, while it concurrently optimizes $L_{\text{PPO}}$ and $L_{\text{dis}}$ differently from InDA adopting the time-separation of optimizing $L_{\text{PPO}}$ and $L_{\text{dis}}$. We also note that it differs from the original method proposed in [36] alternating the main and auxiliary tasks to accomplish every task at equal priority, while we have a clear priority on RL task.

## 5 Experiment

### 5.1 Setups

**Train and test tasks.** We use the OpenAI Procgen benchmark of 16 video games [5], where a main character tries to achieve a specific goal, e.g., finding exit (Maze) or collecting coins (Coinrun), while avoiding enemies given a 2D map. At each time $t$, visual observation $o_t$ is given as an image of size $64 \times 64$. A train or test task is to achieve a high score on a set of environments configured by game and mode, where a mode describes predefined sets of levels (e.g., complexity of map) and backgrounds. Cobbe *et al.* [5] provide *easy* mode for each game, consisting of 200 levels and a certain set of backgrounds.

We simplify *easy* mode and train agents in *easybg* mode, of which the only difference from *easy* mode [5] is showing only a single background. This is useful to investigate the case that using visual augmentation is helpful for testing but not for training. We denote the task configuration by game_name(*mode*), e.g., Coinrun(*easybg*). Then, we evaluate generalization capabilities using two modes: *test-bg* and *test-lv*, which contain unseen backgrounds and levels, respectively, in addition to *easybg* mode that we use for training.

**Types of augmentation.** For clarity, we mainly focus on two types of visual augmentation, each of which conveys distinguishing inductive bias:

(a) *Random color* transforms an image by passing through either color jitter layer or random convolutional layer proposed in [21]. From this, we can impose the consistency to color changes, which may provide a strong generalization to diverse backgrounds of *test-bg* mode.

(b) *Random crop* leaves a randomly-selected rectangle and pads zeros to the rest of the image [25]. This augmentation is particularly useful in fully-observable games (e.g., Chaser and Heist), because it imposes an efficient attention mechanism.

We also report the result with other augmentations including *color jitter, random convolution, gray, and cutout color* in the supplementary material, from which the same messages can be interpreted.

**Baseline methods for RL with data augmentation.** We mainly compare the proposed methods (InDA and ExDA) to the following baselines:

(a) *RAD* [20] simply feeds PPO with experiences of original and augmented observations.

(b) *DrAC* [25] is a method to simultaneously minimizing $L_{\mathrm{PPO}}$ in Eq (1) and $L_{\mathrm{dis}}$ in Eq (4).

(c) *DrAC+PAGrad* is a variant of DrAC, which we devise to investigate another mechanism to relieve the interference between RL training and augmentation in Section 4.5.

The supplementary material presents further details and experiments, which we omitted for simplicity. All results in the main paper are averaged over five runs.

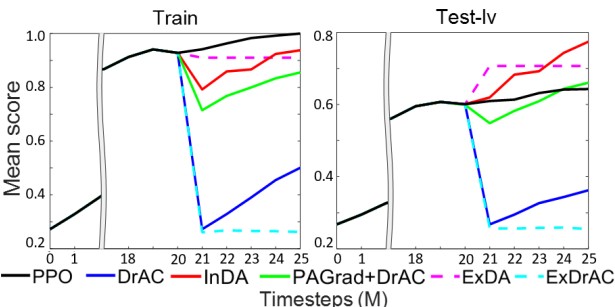

Figure 4: *Benefit of separating distillation from RL training.* We compare InDA, ExDA, DrAC, DrAC+PAGrad and ExDrAC when we start to apply each of them to distill the prior of random crop after 20M timesteps of PPO. ExDrAC is a variant of DrAC without RL training, i.e., minimizing only $L_{\mathrm{dis}}$ in Eq (4).

## 5.2 Benefit of separating distillation from RL training.

In Figure 4, after training PPO agent up to 20M timesteps, we start to suddenly apply one of the different distillation methods with random crop. We report averaged scores over 6 environments (Bigfish, Dodgeball, Plunder, Chaser, Heist, Maze) after normalized by the highest train score of PPO on each environment. After 20M timesteps, ExDA and ExDrAC have no RL training but minimize $L_{\mathrm{DA}}$ in Eq (5) and $L_{\mathrm{dis}}$ in Eq (4), respectively. The substantial gap between ExDA and ExDrAC justifies the design of DA explicitly preserving the knowledge from RL when distilling the prior. More importantly, it is remarkable that ExDA promptly learns the generalization ability once it starts to distill the prior. This validates the idea of postponing the distillation after RL training.

We now compare the distillation methods (InDA, DrAC, and DrAC+PAGrad) concurrently optimizing the RL objective and distilling the prior in Figure 4. Each method has performance degeneration due to the interference generated by distillation. However, InDA and DrAC+PAGrad have clearly smaller degeneration than DrAC which is the only one without any separation of optimizing the RL objective and distillation loss. We note that DrAC+PAGrad has more interference than InDA, and it seems to fail to impose the prior since there is not much difference to PPO in testing. Hence, this verifies the superiority of InDA which enables distilling the prior while alleviating the interference.

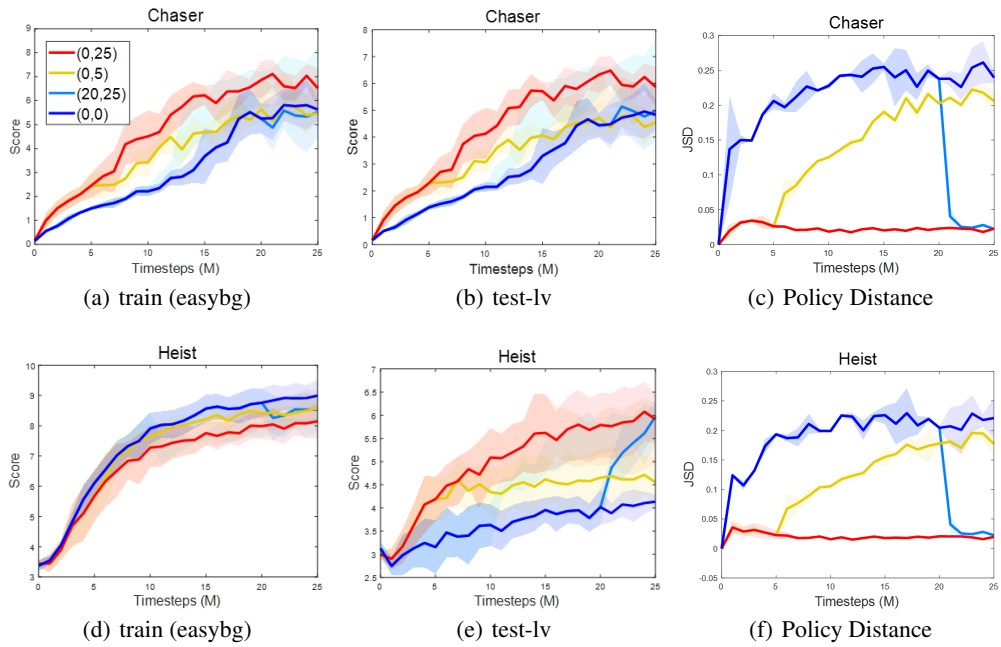

(a) train (easybg)  (b) test-lv  (c) Policy Distance

(d) train (easybg)  (e) test-lv  (f) Policy Distance

Figure 5: *Time-sensitivity of applying augmentation.* We compare train and test performance of InDA$[S, T]$ with *random crop*, where the timing of applying DA is governed by starting time $S$ and terminating time $T$, and we evaluate four different pairs of $[S, T] = [0, 0], [0, 5], [20, 25], [0, 25]$ up to 25M timesteps. Furthermore, we show the change of distance between two policies on an original observation and an augmented observation. Note that InDA with $[S, T] = [0, 0]$ means RL training with no augmentation, i.e., vanilla PPO. We focus on Chaser and Heist since they exemplify two representative time-dependencies. Each train task uses *easybg* mode. We present further details and results with other games in the supplementary material.

## 5.3 Effective timings of distillation

In what follows, we aim to identifying effective timings of distillation. To this end, in Figure 5, we test several schedules of applying DA on two representative environments of distinguishing traits. The supplementary material presents the result on more environments, which is similar one of the representatives.

**An effective timing: InDA.**  Chaser with random crop (Figure 5(a) and 5(b)) represents the case when augmentation improves the sample efficiency of training and thus the generalization ability in training. To compare the generalization ability, we measure policy distance between original and augmented observations using Jenson-Shannon divergence (Figure 5(c)). InDA[0, 5]'s policy distance is increased after it stops using DA. Thus, the generalization ability is degraded, if we do not continue to use DA. In this case, it is clear that InDA should be applied during the entire RL training, as InDA[0, 25] shows the best performance in both training and testing. In addition, it is also important to apply DA as soon as possible since the effect of InDA[0,5] applying DA in the beginning is relatively prompt and significant comparing to that of InDA[20,25]. This suggests that the automatic framework needs to explore more in the early stage.

**An effective timing: ExDA.**  Heist with *random crop* (Figure 5(d) and 5(e)) shows the opposite use of data augmentation to what Chaser case suggests, i.e., postponing data augmentation as much as possible. Random crop generates a slight interference, although it immediately improves the generalization ability. We remark that the inductive bias from the random crop is easily forgotten, as the test performance of InDA[0, 5] is saturated right after stopping the distillation. This can be explained by the time-varying nature of sample distribution and objective in RL. Interestingly, it is in contrast to the data augmentation in SL, where an early distillation is sufficient to impose the prior [10]. On the other hand, the test performance curve of InDA[20, 25] soars right after DA. Furthermore, Figure 5(f) shows that InDA[20, 25] narrows the gap between two policies on the

original and augmented observation. Recalling the interference between RL training and distillation, this suggests postponing the distillation at the end of RL training, and motivates our ExDA.

**Performance benchmark on InDA and ExDA.** In Table 1, we summarize the train and test performances of InDA and ExDA on a set of games with different augmentations and modes. ExDA outperforms other baselines with *random color* on *test-bg*. Moreover, we note that ExDA consumes only 0.5M timesteps to inject the prior at the end of RL training, whereas the others use all the training data. ExDA in both sample efficiency and generalization ability with *random crop* on *test-lv*. It is worth noting that DrAC+PAGrad is slightly better than DrAC, while there is a substantial gap between InDA and each DrAC-based algorithm. This again confirms the benefit from the separated distillation of InDA observed in Figure 4. These results demonstrate that each combination of environment and augmentation has a suitable time at which to apply augmentation, and the gain from using the right distillation timing, i.e., online (InDA, DrAC, or DrAC+PAGrad) or offline (ExDA), is rigid regardless of the choice of algorithms.

| Augmentation | | Task | PPO | RAD | DrAC | DrAC+PAGrad | InDA | ExDA |
|---|---|---|---|---|---|---|---|---|
| Rand color | Rand conv | Train | **1.00** | 0.98 | 0.88 | 0.89 | 0.88 | 0.98 |
| | | Test-bg | 1.00 | 1.08 | 1.86 | 1.88 | 1.92 | **2.11** |
| | | Test-lv | **1.00** | 0.81 | 0.84 | 0.84 | 0.7 | 0.87 |
| | Color jitter | Train | **1.00** | 0.94 | 0.95 | 0.95 | 0.96 | 0.98 |
| | | Test-bg | 1.00 | 1.37 | 1.44 | 1.44 | 1.43 | **1.48** |
| | | Test-lv | **1.00** | 0.83 | 0.86 | 0.86 | 0.84 | 0.88 |
| Rand crop | | Train | 1.00 | 0.28 | 1.08 | 1.09 | **1.25** | 0.91 |
| | | Test-bg | **1.00** | 0.64 | 0.87 | 0.94 | 0.94 | 0.95 |
| | | Test-lv | 1.00 | 0.46 | 1.52 | 1.53 | **1.80** | 1.09 |

Table 1: *Benchmark of InDA and ExDA.* We report normalized train and test scores of InDA, ExDA and DrAC with PAGrad on Open AI Procgen, compared to baselines PPO, DrAC [25], RAD [20]. **Boldface** indicates the best performance. We average the score among several environments [(Rand color: coinrun, ninja, climber, fruitbot, jumper, heist, maze), (Rand crop: Bigfish, Dodgeball, Plunder, Heist, Chaser, Maze)] after normalization considering PPO score to be 1. Every method is trained on 200 levels, using *easybg* mode. We evaluate test performance on both *test-bg* and *test-lv*. The results can be interpreted as an upper bound of potential gain from using data augmentation at perfect timing.

## 5.4 Adaptive scheduling methods

**Adaptive selecting of timings: UCB-InDA and UCB-ExDA.** It is hard to know in advance whether a certain augmentation helps the training or not [25]. We hence employ UCB-InDA which estimates the gain or damage from each augmentation from trial and error and identifies the one

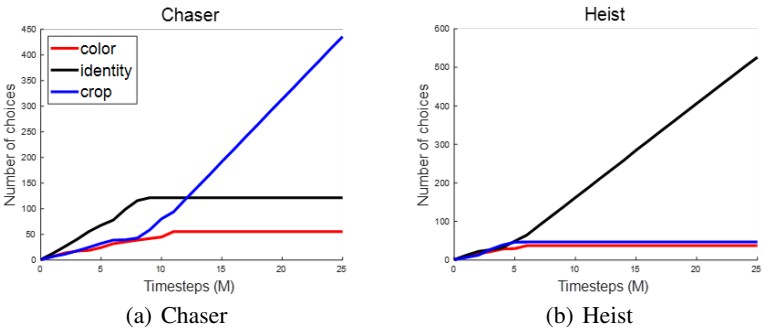

(a) Chaser                      (b) Heist

Figure 6: *Exploration & exploitation to find the most beneficial augmentation.* We show that selected augmentation during training with UCB-InDA for each Chaser and Heist. UCB-InDA automatically selects the augmentation among three options, random color, identity and random crop.

| Env | Method | PPO | DrAC | UCB-DrAC | InDA | ExDA | UCB-InDA | UCB-ExDA |
|---|---|---|---|---|---|---|---|---|
| Heist | Train | 9.2 ± 0.46 | 3.53 ±0.3 | 7.41± 2.09 | 4.9 ± 0.79 | 9.14 ± 0.56 | **9.67± 0.23** | 9.45 ± 0.29 |
| | Test-bg | 5.18 ± 1.53 | 3.58 ± 0.31 | 3.76± 0.54 | 4.9 ± 0.87 | 7.05 ± 1.29 | 6.23± 1.29 | **7.86± 0.83** |
| | Test-lv | 4.13 ± 1.39 | 3.07 ± 0.33 | 3.49±0.48 | 1.47± 0.77 | 5.05 ± 0.98 | 4.8± 1.24 | **5.74± 0.67** |
| Chaser | Train | 5.63± 1.12 | 0.16±0.02 | 4.6± 1.22 | 5.69± 1.42 | 5.58± 1.33 | **7.49±1.27** | 7.23± 1.15 |
| | Test-bg | 0.87± 0.06 | 0.1± 0.02 | 0.57±0.12 | 3.51± 1.33 | 1.02± 0.04 | 1.08± 0.08 | **3. 18± 0.79** |
| | Test-lv | 4.83± 0.88 | 0.14±0.01 | 4.14±1.01 | 4.96± 1.21 | 5.11± 1.05 | **6.45± 0.8** | 6.43± 1.28 |

Table 2: *Full exploitation of augmentation to improve generalization on both test-bg and test-lv*. We compare InDA, ExDA, DrAC, UCB-DrAC, UCB-InDA, UCB-ExDA and PPO about train, test-bg and test-lv. **Boldface** indicates the best method. InDA and DrAC use both random color and random crop during RL training. ExDA use both augmentation after RL training. UCB-InDA and UCB-DrAC are trained as automatically selecting the augmentation during training. UCB-ExDA trains ExDA after UCB-InDA with both augmentation.

with most help. Recall Table 2 where PPO (without augmentation) shows much better training performance than InDA in Heist. As shown in Figure 6(b), UCB-InDA is able to identify that no augmentation is best for training in Heist. This implies that ExDA is more appropriate than InDA. Conversely, random crop is selected on Chaser (Figure 6(a)). As the result, we can automatically select InDA or ExDA appropriately for each task.

**Fully exploitation of augmentation** In Table 2, when both random color and random crop are used to improve generalization on both *test-bg* and *test-lv*, we report numerical evaluation of UCB-InDA and UCB-ExDA with other baselines on train and test tasks. Decreased train performance of DrAC and InDA compare to PPO show the difficulty of simultaneous training with several augmentations. Train performance of UCB-InDA and UCB-DrAC are improved by adaptive selecting, especially, UCB-InDA is better than UCB-DrAC. The gap is made due to the robustness about the change of augmentation during training. In terms of generalization, UCB-ExDA clearly surpasses UCB-InDA thanks to ExDA to extract all the priors from the complete set of data augmentations at the end of RL training.

## 6   Discussion

We have identified two most effective yet simple timings (InDA and ExDA) of data augmentation for RL, and proposed UCB-ExDA framework to adaptively select the best scheduling augmentations. We note that the effectiveness of this framework is restricted but specialized for RL with the unique non-stationary nature. Indeed, in SL without shift of data distribution and objective, it is sufficient to apply data augmentation at the beginning [10]. Our framework employs the most basic multi-armed bandit algorithm with a finite set of data augmentation. It is interesting to investigate a room to improve by further optimizing continuous parameters of data augmentation for RL, c.f., an auto-augmentation technique to optimize continuous augmentation parameter per sample for SL [11]. Another promising direction is to accelerate the distillation process of DA by data condensation with augmentation [37]. This is possible with our framework clearly separating between RL training and distillation, and may be particularly useful to train distributed RL agents since a condensed data for an agent's distillation is usable for the other.

## Acknowledgments and Disclosure of Funding

We thank Kimin Lee for helpful discussions. This work was supported by Institute of Information & communications Technology Planning & Evaluation (IITP) grant funded by the Korea government (MSIT) (No.2019-0-01906, Artificial Intelligence Graduate School Program (POSTECH)) and Institute of Information & communications Technology Planning & Evaluation (IITP) grant funded by the Korea government (MSIT) (No.2021-0-02068, Artificial Intelligence Innovation Hub) and the National Research Foundation of Korea (NRF) grant funded by the Korea government (MSIT) (No. 2021M3E5D2A01023887). Byungchan Ko was supported by the Institute of Information & Communications Technology Planning & Evaluation (IITP) grant funded by Korea (MSIT) (2020-0-01594, PSAI industry-academic joint research and education program).

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
