| $9.2 \pm 0.46$ | $3.53 \pm 0.3$ | $7.41 \pm 2.09$ | $4.9 \pm 0.79$ | $9.14 \pm 0.56$ | $\mathbf{9.67 \pm 0.23}$ | $9.45 \pm 0.29$ |
| | Test-bg | $5.18 \pm 1.53$ | $3.58 \pm 0.31$ | $3.76 \pm 0.54$ | $4.9 \pm 0.87$ | $7.05 \pm 1.29$ | $6.23 \pm 1.29$ | $\mathbf{7.86 \pm 0.83}$ |
| | Test-lv | $4.13 \pm 1.39$ | $3.07 \pm 0.33$ | $3.49 \pm 0.48$ | $1.47 \pm 0.77$ | $5.05 \pm 0.98$ | $4.8 \pm 1.24$ | $\mathbf{5.74 \pm 0.67}$ |
| Chaser | Train | $5.63 \pm 1.12$ | $0.16 \pm 0.02$ | $4.6 \pm 1.22$ | $5.69 \pm 1.42$ | $5.58 \pm 1.33$ | $\mathbf{7.49 \pm 1.27}$ | $7.23 \pm 1.15$ |
| | Test-bg | $0.87 \pm 0.06$ | $0.1 \pm 0.02$ | $0.57 \pm 0.12$ | $3.51 \pm 1.33$ | $1.02 \pm 0.04$ | $1.08 \pm 0.08$ | $\mathbf{3.18 \pm 0.79}$ |
| | Test-lv | $4.83 \pm 0.88$ | $0.14 \pm 0.01$ | $4.14 \pm 1.01$ | $4.96 \pm 1.21$ | $5.11 \pm 1.05$ | $\mathbf{6.45 \pm 0.8}$ | $6.43 \pm 1.28$ |

Table 2: *Full exploitation of augmentation to improve generalization on both test-bg and test-lv*. We compare InDA, ExDA, DrAC, UCB-DrAC, UCB-InDA, UCB-ExDA and PPO about train, test-bg and test-lv. **Boldface** indicates the best method. InDA and DrAC use both random color and random crop during RL training. ExDA use both augmentation after RL training. UCB-InDA and UCB-DrAC are trained as automatically selecting the augmentation during training. UCB-ExDA trains ExDA after UCB-InDA with both augmentation.

with most help. Recall Table 2 where PPO (without augmentation) shows much better training performance than InDA in Heist. As shown in Figure 6(b), UCB-InDA is able to identify that no augmentation is best for training in Heist. This implies that ExDA is more appropriate than InDA. Conversely, random crop is selected on Chaser (Figure 6(a)). As the result, we can automatically select InDA or ExDA appropriately for each task.

**Fully exploitation of augmentation** In Table 2, when both random color and random crop are used to improve generalization on both *test-bg* and *test-lv*, we report numerical evaluation of UCB-InDA and UCB-ExDA with other baselines on train and test tasks. Decreased train performance of DrAC and InDA compare to PPO show the difficulty of simultaneous training with several augmentations. Train performance of UCB-InDA and UCB-DrAC are improved by adaptive selecting, especially, UCB-InDA is better than UCB-DrAC. The gap is made due to the robustness about the change of augmentation during training. In terms of generalization, UCB-ExDA clearly surpasses UCB-InDA thanks to ExDA to extract all the priors from the complete set of data augmentations at the end of RL training.

# 6   Discussion

We have identified two most effective yet simple timings (InDA and ExDA) of data augmentation for RL, and proposed UCB-ExDA framework to adaptively select the best scheduling augmentations. We note that the effectiveness of this framework is restricted but specialized for RL with the unique non-stationary nature. Indeed, in SL without shift of data distribution and objective, it is sufficient to apply data augmentation at the beginning [11]. Our framework employs the most basic multi-armed bandit algorithm with a finite set of data augmentation. It is interesting to investigate a room to improve by further optimizing continuous parameters of data augmentation for RL, c.f., an auto-augmentation technique to optimize continuous augmentation parameter per sample for SL [12]. Another promising direction is to accelerate the distillation process of DA by data condensation with augmentation [39]. This is possible with our framework clearly separating between RL training and distillation, and may be particularly useful to train distributed RL agents since a condensed data for an agent's distillation is usable for the other.

# Acknowledgments and Disclosure of Funding

We thank Kimin Lee for helpful discussions. This work was supported by Institute of Information & communications Technology Planning & Evaluation (IITP) grant funded by the Korea government (MSIT) (No.2019-0-01906, Artificial Intelligence Graduate School Program (POSTECH)) and Institute of Information & communications Technology Planning & Evaluation (IITP) grant funded by the Korea government (MSIT) (No.2021-0-02068, Artificial Intelligence Innovation Hub) and the National Research Foundation of Korea (NRF) grant funded by the Korea government (MSIT) (No. 2021M3E5D2A01023887). Byungchan Ko was supported by the Institute of Information & Communications Technology Planning & Evaluation (IITP) grant funded by Korea (MSIT) (2020-0-01594, PSAI industry-academic joint research and education program).

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

# A  Modified Procgen Environments

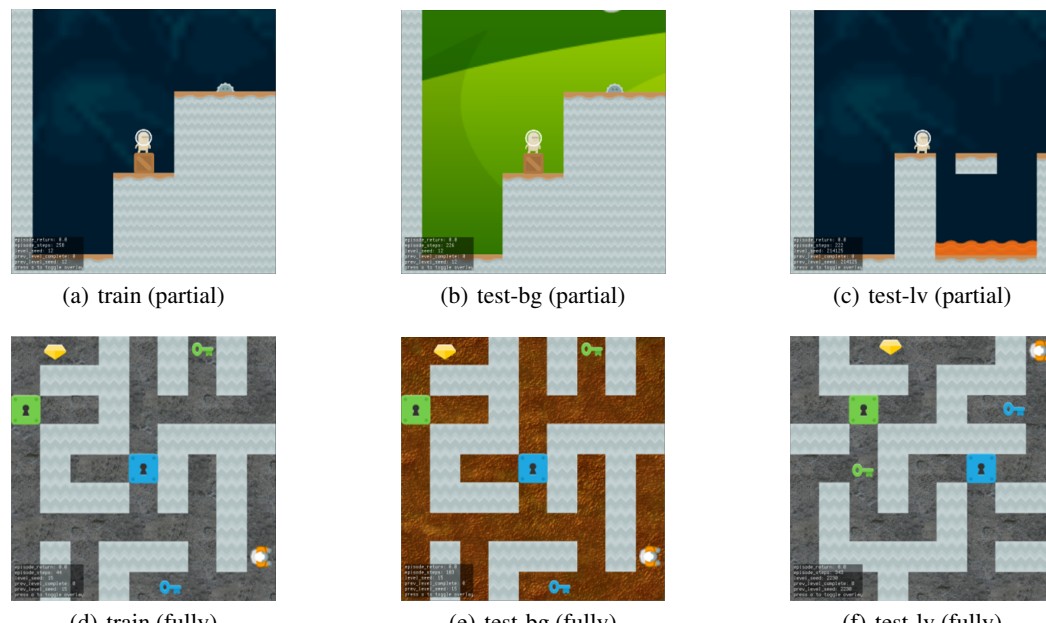

|  |  |  |
| --- | --- | --- |
| (a) train (partial) | (b) test-bg (partial) | (c) test-lv (partial) |
| (d) train (fully) | (e) test-bg (fully) | (f) test-lv (fully) |

Figure 7: An example set of training and testing environments in Procgen benchmark: (upper row) an example of partially observable environment with Coinrun; (lower row) an example of fully observable environment with Heist; (left column) train: a set of levels and backgrounds for training; (center column) test-bg: the same training levels on unseen backgrounds; (right column) test-lv: a set of unseen levels on the same training backgrounds

This section explains Modified Procgen Environments, which is designed to verify different types of generalization, backgrounds, and levels. Open AI Procgen environments [5] share background themes such as *space_backgrounds*, *platform_backgrounds*, *topdown_backgrounds*, *water_backgrounds*, *water_surface_backgrounds*. We create new difficulties as *Easybg*, *Easybg-test*, *Easy-test*. *Easybg* generates environments which contain only one for each background, wall and agent theme. *Easybg-test* and *Easy-test* are for test about background change after trained on *Easybg* and *Easy*. Wall theme in (Climber, Coinrun, Jumper, Ninja) and Agent theme in (Climber, Coinrun) also compose with only one image resource in *Easybg*. Figure 7 presents an example set of modes that we use in evaluation. Furthermore, We fix the exit_wall_choice and enemy theme in Dodgeball. We describe the usage themes in each environment, which are grouped by backgrounds theme as below:

- *space_backgrounds* (Bossfight, Starpilot)
  Background: "space_backgrounds/deep_space_01.png"
- *platform_backgrounds* (Caveflyer, Climber, Coinrun, Jumper, Miner, Ninja)
  Background:"platform_backgrounds/alien_bg.png", Coinrun (Agent color: Beige, Wall themes: Dirt), Climber (Agent color: Blue, Wall themes: tileBlue), Jumper (Wall theme: tileBlue), Ninja (Wall theme: bricksGrey)
- *topdown_backgrounds* (Chaser, Dodgeball, Fruitbot, Heist, Leaper, Maze)
  Background:"topdown_backgrounds/floortiles.png", Dodgeball (Enemy theme: "misc_assets/character1.png", Exit_wall_choice: 0)
- *water_backgrounds* (Bigfish)
  Background:"water_backgrounds/water1.png"
- *water_surface_backgrounds* (Plunder)
  Background:"water_backgrounds/water1.png"

*Easybg-test* uses backgrounds in each background group, except the one used in *Easybg*. *Easy-test* is only defined for Climber, Jumper, Ninja, and they compose with *topdown_backgrounds*.

# B  Implementation details

In this section, we explain about InDA, ExDA, UCB-InDA and other baselines. We train the agent with IMPALA-CNN [8] in every experiment.

## B.1  InDA

We use PPO [29] as a base RL algorithm, For data efficiency, we store the observations during RL training in buffer $\mathcal{D}_O$. Before DA phase, we also make policy buffer $\mathcal{D}_\Pi$, value function buffer $\mathcal{D}_V$ and augmented observation buffer $\mathcal{D}_\phi$ for distillation, because we only use one network model. We randomly sample pairs of $(o, \pi, V)$ from buffer, and minimize loss function $L_{DA}(\theta)$. We reuse the sample three times like PPO, it can be controlled by # Epochs of DA. We did a greed searches for learning rate of DA $l_{DA} \in [1 \times 10^{-3}, 5 \times 10^{-4}, 2 \times 10^{-4}, 1 \times 10^{-4}, 5 \times 10^{-5}]$ and interval $I \in [1, 5, 10]$ and found the best combination $l_{DA} = 10^{-4}$ and interval $I = 5$. We fix the buffer size $\mathcal{D}_O = 40960$, because we collect the observations during five RL phases ($5 \times 256 \times 32$). We describe the every hyperparameter as below:

| Hyperparamter | Value |
|---|---|
| $\gamma$ | 0.999 |
| $\lambda$ | 0.95 |
| # Timesteps per rollout | 256 |
| # Epochs per rollout | 3 |
| # Minibatches per epoch | 8 |
| Reward Normalization | Yes |
| # Workers | 1 |
| # Environments per worker | 32 |
| Total timesteps | 25M |
| LSTM | No |
| Frame Stack | No |
| Optimizer | Adam optimizer |
| Entropy bonus | 0.01 |
| PPO clip range | 0.2 |
| Learning rate | $5 \times 10^{-4}$ |
| Interval $I$ | 5 |
| Size of $\mathcal{D}_O$ | 40960 |
| # Epochs of DA | 3 |
| Learning rate of DA $l_{\mathrm{DA}}$ | $1 \times 10^{-4}$ |
| Image transformation $\phi$ | Any augmentation |

## B.2  ExDA

In ExDA, we generate and store $(o, \pi, V)$ using $f_{\theta_{old}}$ in buffer $\mathcal{D}$. The optimal buffer size depends on the episode length of each environment. However, we standardize the buffer size as 0.5M in every environment. We augment the observation with three epochs intervals when using randomized augmentation methods. We did greed searches for # minibatches $[1024, 2048, 4096]$ and learning rate $[5 \times 10^{-4}, 1 \times 10^{-3}, 2 \times 10^{-3}]$. As a result, we select # of minibatches 4096 and a learning rate $1e-3$. We describe every hyperparameter as below:

| Hyperparameter | Value |
|---|---|
| Size of $\mathcal{D}_O$ | 0.5M |
| # Epoch | 30 |
| # Minibatches per epoch | 4096 |
| Learning rate | $1 \times 10^{-3}$ |
| # Workers | 1 |
| Optimizer | Adam optimizer |
| Image transformation $\phi$ | Any augmentation |

## B.3 UCB-InDA

We use UCB-InDA as a discriminator to determine the necessity of augmentation during the training. The gain of an augmentation is a mean of return during interval I, $G(s) = \frac{1}{I} \sum_{i=0}^{j-1} R(s+j)$. The return is computed by the sum of estimated advantage and predicted value, which are expected value of the agent trajectory, $R(s) = \hat{\mathbb{E}}_{(o_t, a_t) \sim \pi_\theta}[\hat{A}_t + V_\theta(o_t)]$. The $\hat{A}_t$ is advantage from Generalized Advantage Estimator [28]. Thus, we can evaluate how augmentation affects the return on the agent trajectory. However, the distribution of return is non-stationary, as the agent policy is changed. Therefore, we use the window average gain $\bar{G}_\phi$ rather than the whole gain from the past evaluation. Furthermore, the drastic change of return causes the gap of gain between the augmentation according to sampling time at the transient time of training and leads to poor exploration about some augmentation methods. For stable exploration, we fix the minimum exploration frequency and use forced exploration method after the minimum exploration as below:

$$\bar{G}_{\phi_{max}}(s) + c\sqrt{\frac{\log(s)}{N_{\phi_{max}}(s)}} \leq \bar{G}_{\phi_{min}}(s) + c\sqrt{\frac{\log(s)}{N_{\phi_{min}}(s)}} \tag{9}$$

$$c = \frac{\bar{G}_{\phi_{max}} - \bar{G}_{\phi_{min}} + \epsilon}{\sqrt{\log(s)} \times \max\left(\frac{1}{\sqrt{N_{\phi_{min}}(s)}} - \frac{1}{\sqrt{N_{\phi_{max}}(s)}}, \frac{1}{\sqrt{W-1}} - \frac{1}{\sqrt{W}}\right)} \tag{10}$$

where $\phi_{max} = \arg\max_{\phi \in \Phi} \bar{G}_\phi$, $\phi_{min} = \arg\min_{\phi \in \Phi} \bar{G}_\phi$. We set the hyperparameter as below table:

| Hyperparameter | Value |
|---|---|
| Window size of gain W | 3 |
| Minimum exploration frequency | 15 |

## B.4 Baselines

We compare ExDA and InDA with PPO [5], DrAC [26], Rand-FM [22], RAD [21]. Every baseline is based on PPO [5] and we adopt the implementation of PPO in [5].

- DrAC [26] regularizes both policy and value function as self-supervised learning. Regularization term have hyperparameter $\alpha_r$ for ratio with PPO objective. We use the hyperparameter recommended by the author.

- Rand-FM [22] is composed with random convolution networks and feature matching. They also need hyperparameter $\beta$ for ratio between feature matching and PPO objective. We use same $\beta$ with author.

- RAD [21] naively use augmented observations in state distribution. Thus, there are no additional hyperparameters.

We describe the hyperparameter of baselines in below table:

| Hyperparameter | Value |
|---|---|
| $\gamma$ | 0.999 |
| $\lambda$ | 0.95 |
| # of timesteps per rollout | 256 |
| # of epochs per rollout | 3 |
| # of Minibatches per epoch | 8 |
| Reward Normalization | Yes |
| # of Workers | 1 |
| # of environments per worker | 64 |
| Total timesteps | 25M |
| LSTM | No |
| Frame Stack | No |
| Optimizer | Adam optimizer |
| Entropy bonus | 0.01 |
| PPO clip range | 0.2 |
| Learning rate | $5 \times 10^{-4}$ |
| $\alpha_r$ (DrAC) | 0.1 |
| $\beta$ (Rand-FM) | 0.002 |

## C   Data augmentation

In our experiments, we use five augmentation methods: *crop, grayscale, cutout color, random convolution* and *color jitter*. We refer the implementation of augmentations from Lee *et al.* [22] (*random convolution*), Laskin *et al.* [21] (*cutout color, color jitter*) and Raileanu *et al.* [26] (*grayscale, crop*). We expect the generalization about background change from *random convolution, color jitter, gray, cutout color*. About the change of levels, we use *crop* and *cutout color* for generalization. Examples of data augmentation are represented below:

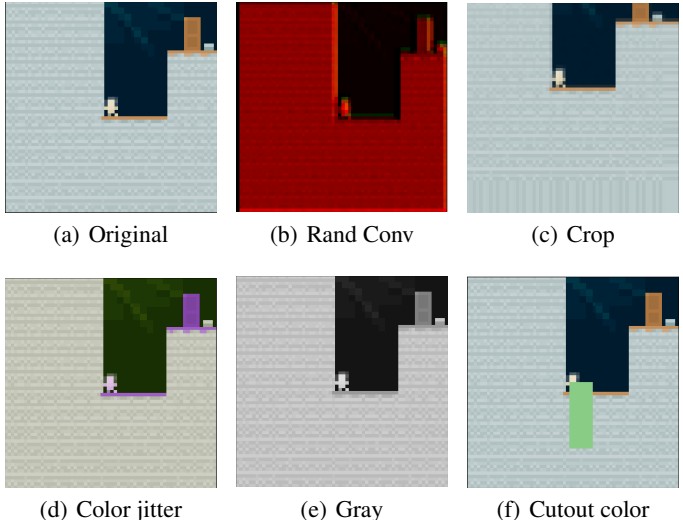

| (a) Original | (b) Rand Conv | (c) Crop |
|---|---|---|
| (d) Color jitter | (e) Gray | (f) Cutout color |

Figure 8: Examples of visual augmentations

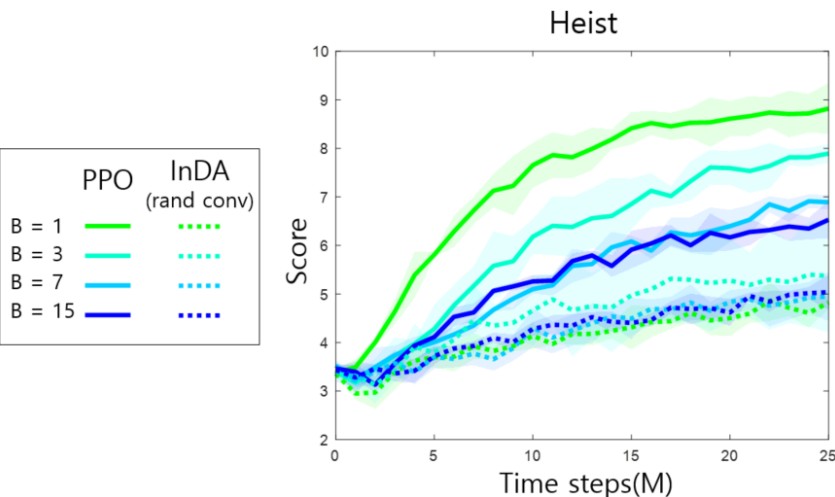

Figure 9: *Interference from increased complexity of learning.* We present train performance when training agent on Heist(*easybg-B*), where *easybg-B* is a variant of *easybg* mode with $B$ backgrounds. We plot the train performance curves of PPO and InDA with random convolution over training epochs in absolute score.

## D  Interference from increased complexity of learning

Even if the policy for the original observation is fixed and augmentation is used, learning may still be hindered. In Figure 9, we report training curves of the agents on train tasks of different degree of background diversity. InDA with random convolution is anticipated to lead the prior on color consistency, which seems helpful to handle background diversity. However, PPO outperforms InDA in terms of the sample complexity to master train tasks, while the gap is decreasing as the background diversity increases.This implies that diverse backgrounds make hard to train by increased complexity of learning, and also data augmentation can cause similar difficulty even with right prior, when using it during RL training. We further remark that ExDA using random convolution after PPO trained on Heist(*easybg-1*) achieves 8.15 on Heist(*easybg-15*), which is much higher than PPO's 6.53 trained on Heist(*easybg-15*). This suggests the importance of simplifying the train task and the utility of ExDA which completely separates RL training and distillation with augmentation.

## E  Robustness in loss function change

In ExDA, we transfer the policy after training 20M time steps with PPO. Thus, we explain why other augmentations are not used after pre-training. We compare the results of training and test performance with Drac [26], Rand-FM [22], Rad [21] when we train each method for 5M after training PPO for 20M time steps. We use random convolution and crop as data augmentation methods, and we do not compare with RAD when we use crop in Figure 12 and Figure 13. The *crop* method used in our paper do not work well in RAD, because they use a different crop method with [26] in their paper [21]. InDA is more stable than others in training, and it affects generalization performance.

Every training curves decline immediately after starting training with augmented observations at 20M time steps. The objective function is changed to each baseline, and augmented data is newly added to data distribution. Thus, the optimizer should find a new optimal point for new objective function and data. During find the new optimal points, the agent learns along with the different directions from the optimization direction in pure PPO. Thus, performance can be degraded because the learning direction on loss landscape is different from maximizing rewards on non-augmented data in PPO. In spite of using self-supervised learning or representation learning, the policy is changed because they update the same network's parameter for matching policy or latent features, such as DrAC [26] and Rand-FM [22]. However, InDA is more stable than the others because we distill the fixed policy and value using DA. It does the stable training through conserving the policy on non-augmented observations during optimizing for augmented data.

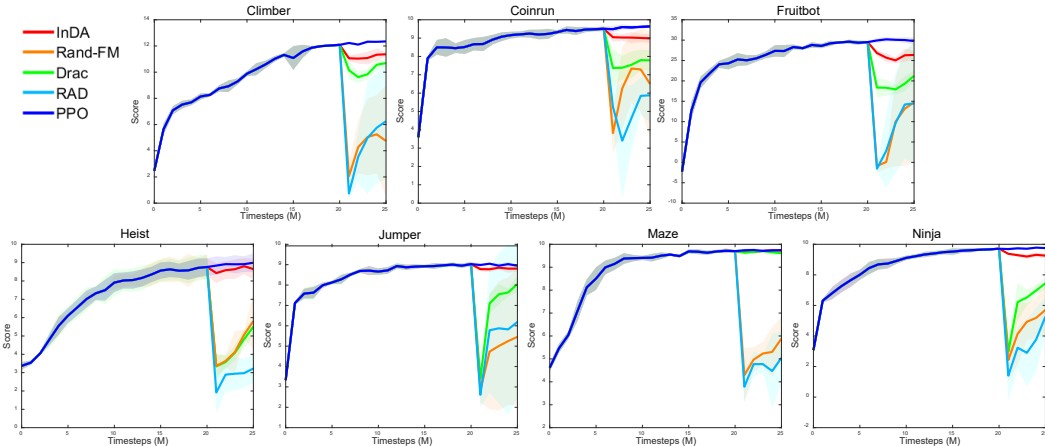

Figure 10: Comparison of the training performance when *random convolution* is applied after 20M timesteps with various augmentation methods.

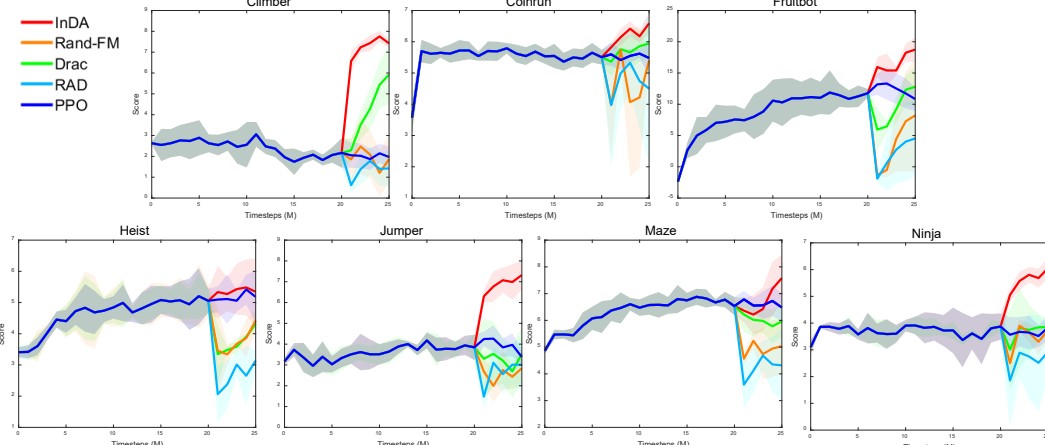

Figure 11: Comparison of the test performance when *random convolution* is applied after 20M timesteps with various augmentation methods.

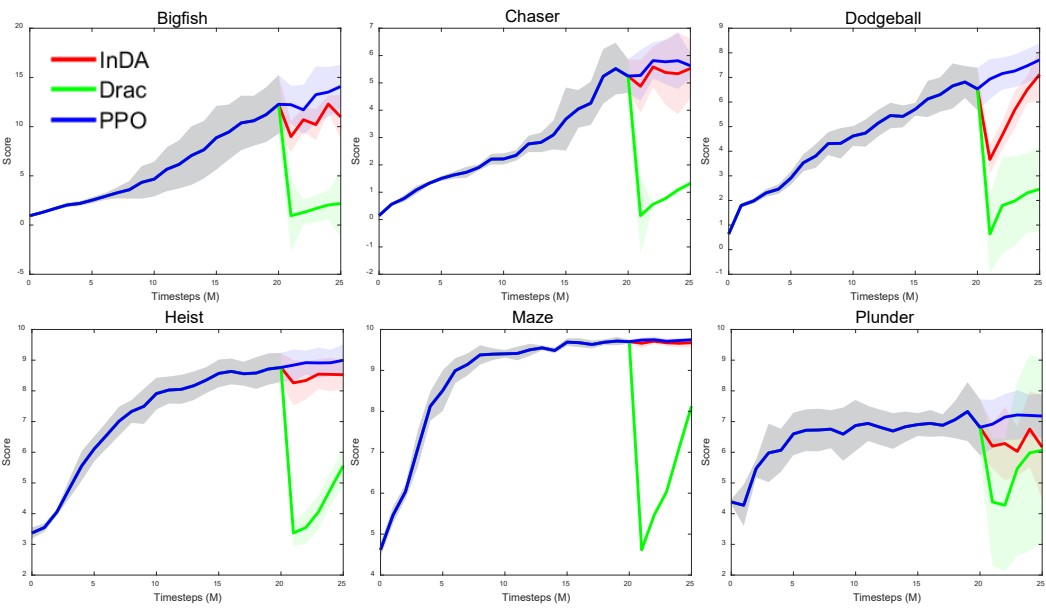

Figure 12: Comparison of the training performance when *crop* is applied after 20M timesteps with InDA and Drac.

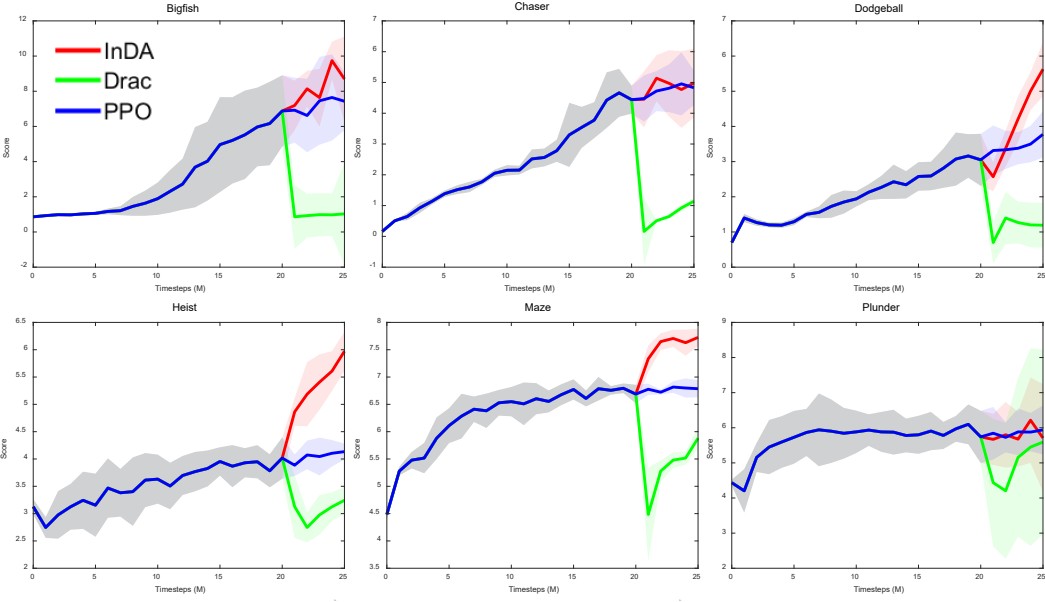

Figure 13: Comparison of the test performance when *crop* is applied after 20M timesteps with InDA and Drac.

## F Robustness against wrong augmentation

In this section, we verify the robustness of ExDA from an obvious wrong augmentation: just making an black image, denoted by *black*. Using this clearly interferes with RL training. We compare PPO, InDA, ExDA, UCB-InDA, UCB-ExDA and DrAC. The hyperparameter is almost the same with Section 5.4 except we use *black* instead of *random color* and *random crop*. For UCB-based methods, we use 4 arms: *black*, *random color*, *random crop*, and *no aug*. As expected, ExDA preserves the PPO score, but InDA and DrAC degrade the score. Furthermore, UCB-InDA improves the performance in

Chaser, and also it almost preserves the score in Heist. Lastly, UCB-ExDA also maintains the score from UCB-InDA.

| Env | PPO | DrAC | InDA | ExDA | UCB-InDA | UCB-ExDA |
|---|---|---|---|---|---|---|
| Heist | **9.2 ± 0.46** | 7.35 ± 0.684 | 6.72 ±0.419 | 8.6 ± 0.189 | 8.84 ± 0.307 | 8.64± 0.264 |
| Chaser | 5.63± 1.12 | 1.49±0.036 | 5.43± 0.633 | 5.1±0.331 | **6.74±0.588** | 6.28± 0.436 |

Table 3: Robustness from the wrong augmentation

# G    Ablation study of ExDA

## G.1    Initialization and regularization term

In this section, we do an ablation study about the factor of ExDA. We mention the loss function and re-initialization issue in subsection 4.3. ExDA does not have to minimize $L_{VD}$ because the value function is useless after RL training. The below results show that $L_{VD}$ cannot give any benefit in ExDA. Thus, we only use $L_{PD}$ for computational complexity. Furthermore, we also compare to verify the effect of non-stationarity with a re-initialized agent before distillation. Igl *et al.* [18] argued that the non-stationarity causes the reduction of generalization. However, the re-initialization is not critical in test performance, as shown in Figure 15. Moreover, sometimes re-initialization makes it difficult to distill training performance such as Fruitbot and Ninja in Figure 14. We use *random convolution* as an augmentation method in here.

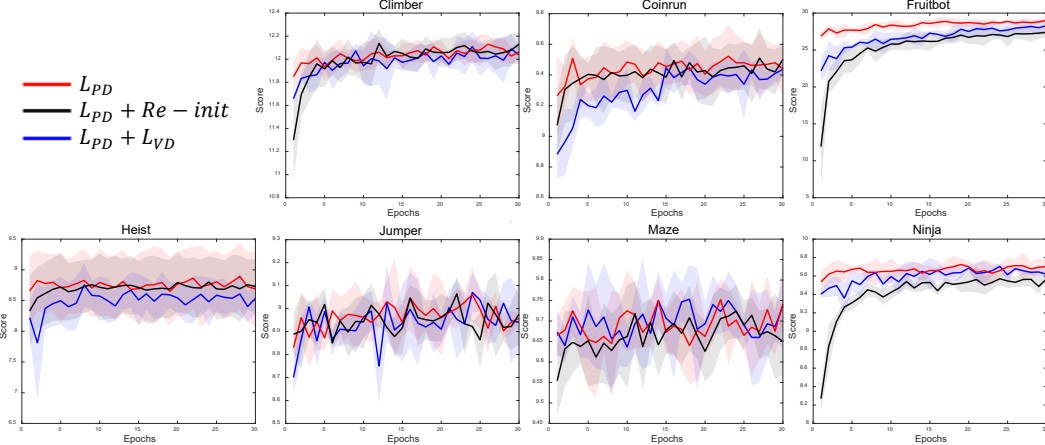

Figure 14: Training performance of ExDA with re-initialization or regularization with value funtion.

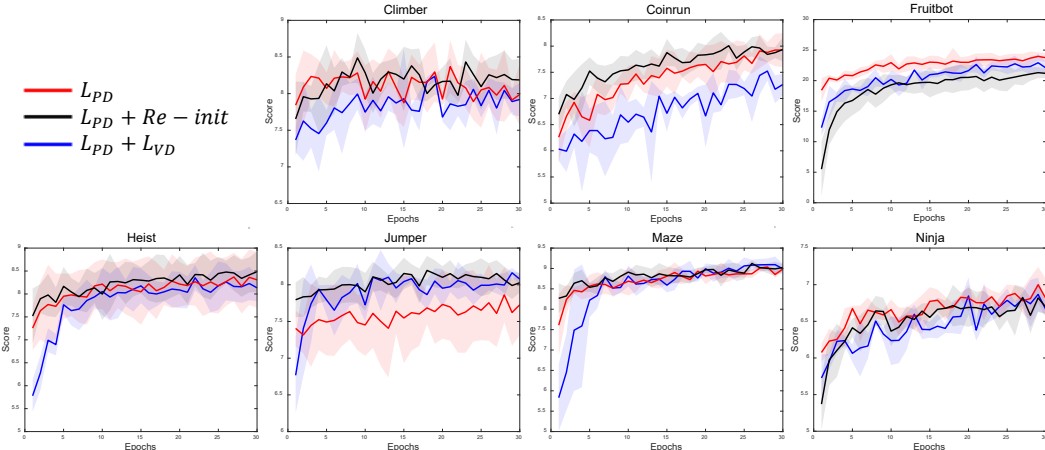

Figure 15: Test performance of ExDA with re-initialization or regularization with value funtion on unseen backgrounds.

## G.2 ExDA after InDA with various backgrounds

When augmentation helps the training, ExDA struggle to follow the training performance of InDA because ExDA's training performance is limited by pre-trained agent's policy. Thus, we use InDA for ExDA's pre-training , and call it as ExDA (InDA). As shown in Table 5, ExDA (InDA) is comparable to InDA, but not beyond. Thus, unless there is a meaningful difference in training performance, ExDA has no better generalization than InDA. However, in computational complexity, ExDA is more efficient than others such as InDA and DrAC when they have a similar performance. In the following section, we discuss computational complexity.

| Easy | PPO | InDA | ExDA (PPO) | ExDA (PPO) + reinit | ExDA (InDA) | ExDA (InDA) + reinit |
|---|---|---|---|---|---|---|
| Jumper | 8.55 | 8.94 | 8.5 | 8.6 | 8.83 | 8.83 |
| | ±0.17 | ±0.09 | ±0.183 | ±0.156 | ±0.215 | ±0.126 |
| Ninja | 7.49 | 8.88 | 7.03 | 7.23 | 8.71 | 8.56 |
| | ±0.42 | ±0.34 | ±0.058 | ±0.159 | ±0.344 | ±0.394 |
| Climber | 8.63 | 8.5 | 8.1 | 8.09 | 8.16 | 7.99 |
| | ±0.46 | ±0.29 | ±0.268 | ±0.268 | ±0.441 | ±0.383 |

Table 4: The comparison with diverse agents which are trained with ExDA

| Easy | PPO | InDA | ExDA (PPO) | ExDA (PPO) + reinit | ExDA (InDA) | ExDA (InDA) + reinit |
|---|---|---|---|---|---|---|
| Jumper | 6.85 | 7.94 | 7.54 | 7.48 | 7.98 | 7.67 |
| | ±0.19 | ±0.19 | ±0.158 | ±0.154 | ±0.148 | ±0.155 |
| Ninja | 6.29 | 6.5 | 5.56 | 5.73 | 6.27 | 5.94 |
| | ±0.19 | ±0.19 | ±0.158 | ±0.154 | ±0.148 | ±0.155 |
| Climber | 6.96 | 7.28 | 7.06 | 6.89 | 6.8 | 5.45 |
| | ±0.65 | ±0.35 | ±0.541 | ±0.237 | ±0.441 | ±0.383 |

Table 5: Test performance of agents, which is trained on easy mode with random convolution.

## G.3 Computational complexity

We compare the computational complexity with ExDA and InDA. InDA do DA for every 25M observations during training and reuse the sample in three times. However, ExDA only use 0.5M for DA during 30 epochs. Thus, ExDA is almost 5 times more efficient than InDA by rough calculation. Furthermore, the ExDA saves the time for augmentation compared to InDA. When we train with same computational setting (GPU: GeForce RTX 2080 TI), ExDA only consumes 5 hours + 2 hours

(PPO) when using random convolution, but, InDA consumes 18 hours. Thus, we recommend ExDA when InDA cannot give a meaningful gain in training performance.

# H  Comparison between InDA and ExDA using the same steps of RL training

In every experiment in the paper, we set the evaluation setup to be somewhat unfavorable to ExDA (using 20M time steps of RL training followed by additional 0.5M steps of distillation; denoted by ExDA(20M)) compared to InDA or other baselines (25M time steps of RL training) to clearly avoid potential complaint about the extra 0.5M steps for ExDA, It is obvious that the performance of ExDA is improved if we put more time steps for RL training, and thus the benefit of ExDA compared to InDA becomes more conspicuous if ExDA is the effective timing. Thus, we do an additional experiment evaluating ExDA (25M) using 25M RL time steps and 0.5M distillation time steps as below table:

| Easybg | | PPO | DrAC | RAD | InDA | ExDA(20M) | ExDA(25M) |
|---|---|---|---|---|---|---|---|
| | Train | **9** | 5.95 | 7.94 | 5.15 | 8.72 | 8.93 |
| Heist | Test-bg | 5.18 | 5.47 | 4.78 | 4.96 | 8.15 | **8.26** |
| | Test-lv | 4.13 | 5.4 | 3.81 | **5.91** | 5.35 | 5.41 |

Table 6:  Additional RL training before ExDA

The result of ExDA(25M) reinforces our main message: postponing data augmentation when it generates a severe interference with RL training in Table 6. ExDA(25M) has a slight drop in train score compared to PPO, but it could be eventually removed if we put (slightly) more steps for distillation.

# I UCB with a large set of augmentation

In Figure 6, we only use three arms such as *random color*, *random crop* and *no augmentation*. Thus, we try to select the usuful augmentations at each time among a large set of augmentation, which contains *gray*, *cutout color*, *random convolution*, *color jitter*, *random crop* and *no augmentation*. In Figure 16(a), we show the result of arm selection with UCB-InDA. The *identity* function is most frequently used in the same as Figure 6. On the other hand, we ablate the neccesity of identity function in UCB-InDA, it shows that the *identity* function is needed.

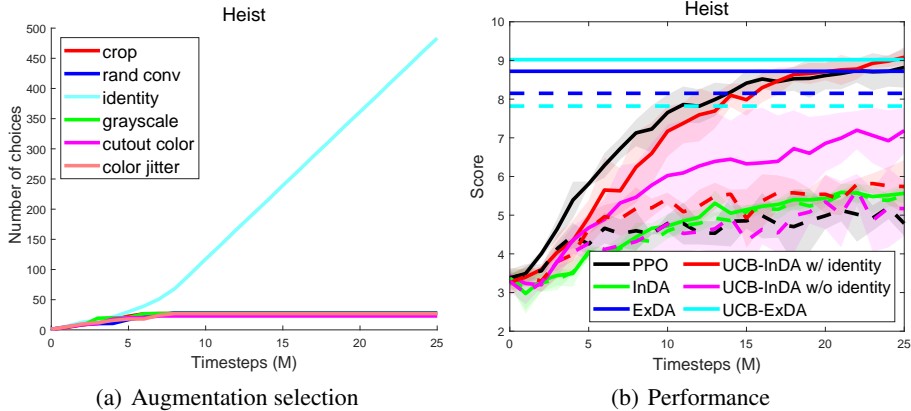

(a) Augmentation selection                    (b) Performance

Figure 16: Figure 16(a) show the selected number of augmentations by UCB on Heist. We compare two UCB-InDAs, w/ and w/o identity function with PPO, InDA, ExDA, UCB-ExDA on Heist *easybg* in Figure 16(b). UCB-InDA is trained after UCB-InDA w/ identity, we use *random convolution* as a data augmentation in InDA, ExDA. Solid line: train performance; dotted line: test performance. ExDA achieves larger test performance than InDA by preserving train performance. Moreover, UCB-InDA w/ identity outperforms UCB-InDA w/o identity in the training.

# J Time matter in training

This section shows every result of Figure 5 about time dependency with InDA. We experiment with *random convolution, crop, color jitter* and evaluate the test on unseen backgrounds (*random convolution, color jitter*) and levels (*random crop*). However, the effect of generalization is hard to recognize in most cases, as shown in Appendix K. Thus, we mainly discuss the most effective augmentation, such as random convolution and crop in the main paper, and only represent some environments that have helped the generalization by color jitter. *easybg* mode is used as default mode with three *easy* mode (Climber, Jumper, Ninja) in our experiments. The shaded regions and solid line represent the standard deviation and mean, across five runs.

## J.1 Random convolution

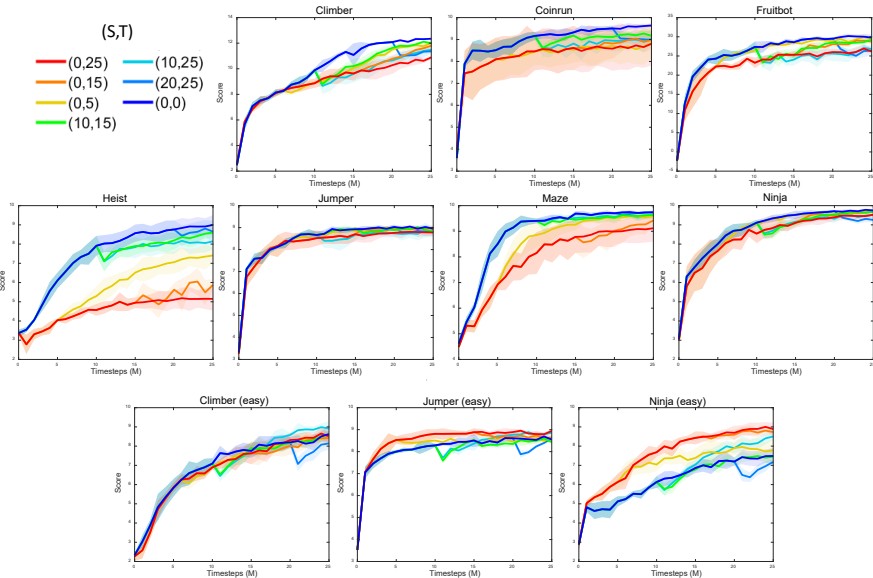

Figure 17: Comparison of training performance according to usage period of augmentation with InDA (*random convolution*): The *easybg* is disturbed by *random convolution*, but, *easy* mode is improved training performance by *random convolution*.

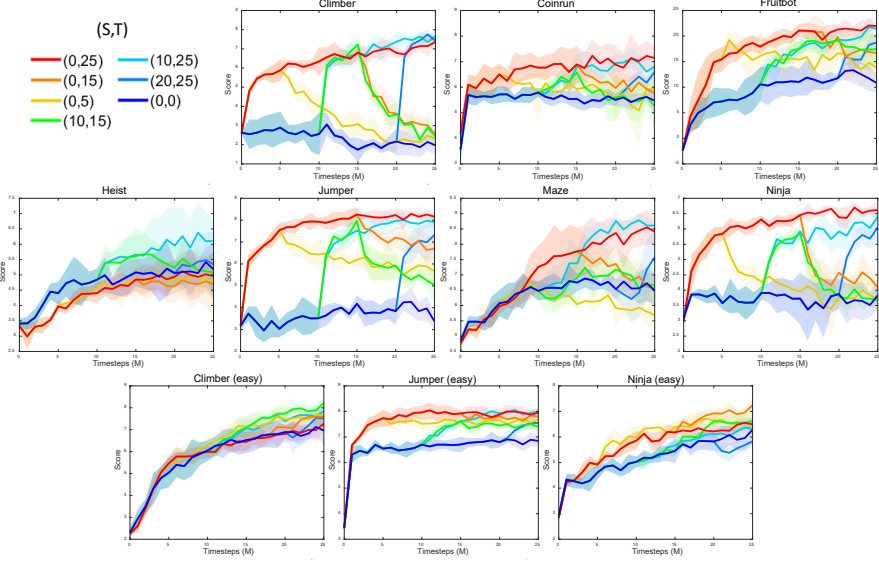

Figure 18: Comparison of generalization on unseen backgrounds according to usage period of augmentation with InDA (*random convolution*): Most cases' tendencies are coincidence with the jumper, which is mentioned in the main paper.

## J.2 Crop

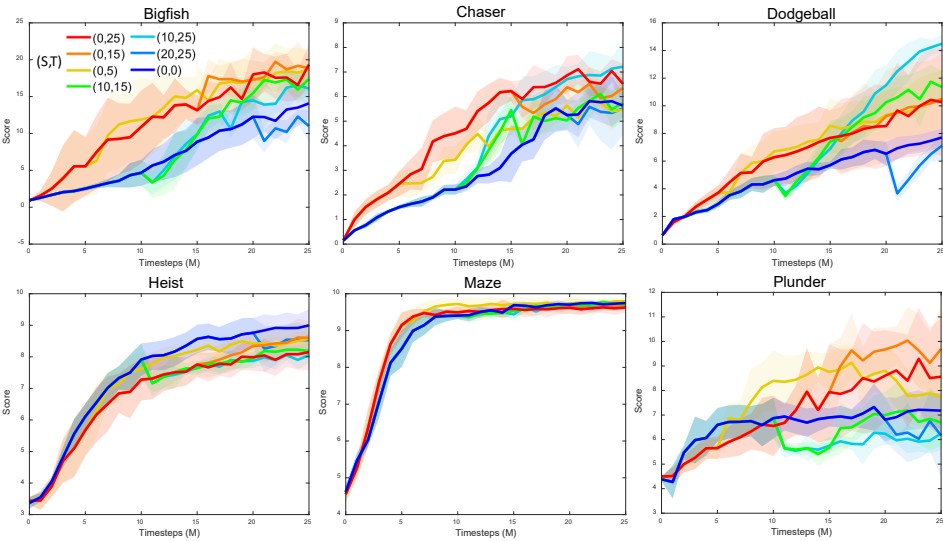

Figure 19: Comparison of training performance according to usage period of augmentation with InDA (*crop*): *Crop* improve the training performance in Bigfish, Chaser, Dodgeball, Plunder. Furthermore, interrupted augmentation is also improved similarly with (0, 25).

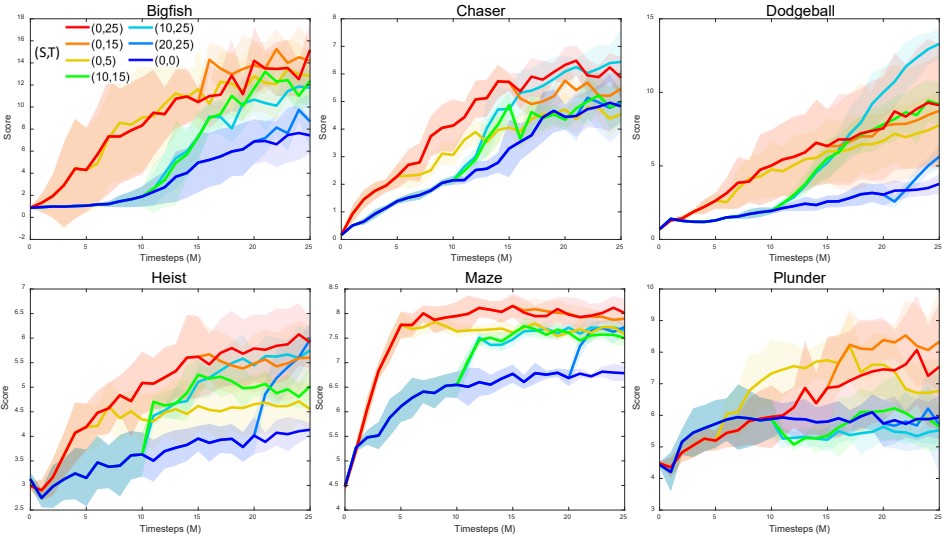

Figure 20: Comparison of generalization on unseen levels according to usage period of augmentation with InDA (*crop*): The generalization is improved by *crop*, and it is conserved after interrupted in Heist and Maze. Bigfish, Chaser, Dodgeball, and Plunder have similar curves with training.

## J.3 Color jitter

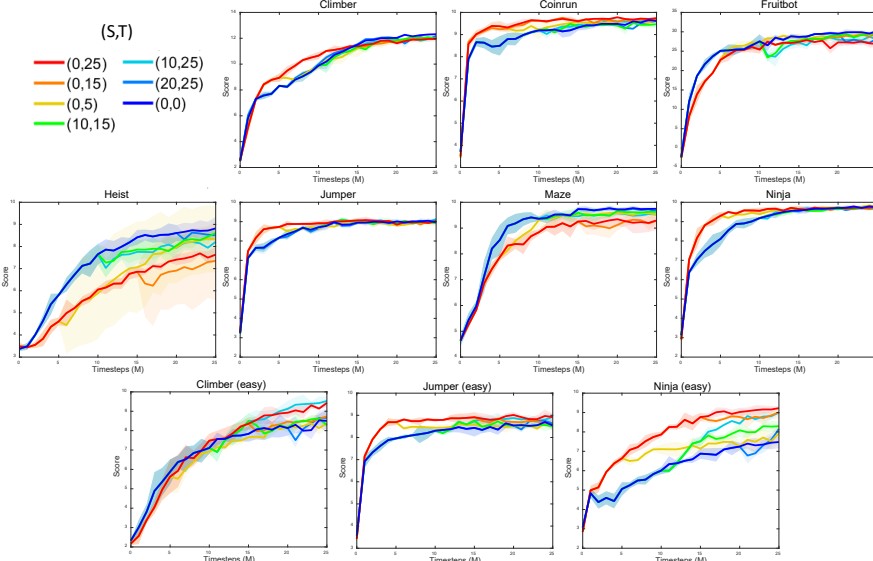

Figure 21: Comparison of training performance according to usage period of augmentation with InDA (*color jitter*): *Color jitter* does not impede the training as much as *random convolution* in most environments. However, *color jitter* helps the training in *easy* mode.

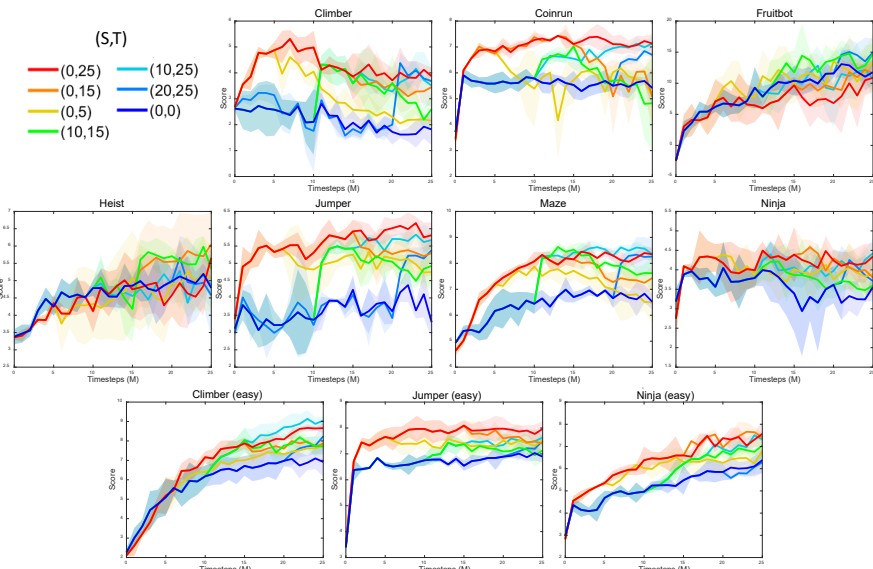

Figure 22: Comparison of generalization on unseen backgrounds according to usage period of augmentation with InDA (*color jitter*): Test performance is influenced by *color jitter* as the trend, which is similar to *random convolution*.

## K   Benchmark on Modified Open AI Procgen

We compare the training and test performance on various environments with each augmentation. We also use DrAC [26], RAD [21], DrAC+PAGrad as baselines. In every result, we train the agent for 25M timesteps, except the ExDA. ExDA is trained with 0.5M after training 20M with PPO. We also compare the average score after normalized by PPO's score and indicate the best score as bold. Mean

and standard deviation is calculated after five runs. We show the result about *random conv, color jitter, random crop* in Table 1. Additionally, we evaluate benchmark with *gray* and *cutout color* in Table 7. For benchmark, we classify Procgen environments with each characteristic as Figure 23. Furthermore, we attach detail results on each environments with Oracle and Rand-FM [22]. Red one is the Oracle score, which is trained on test environments such as *easybg-test, easy-test*. For your information, RAD does not work well when using crop, because we use [26]'s crop method which is different with [21].

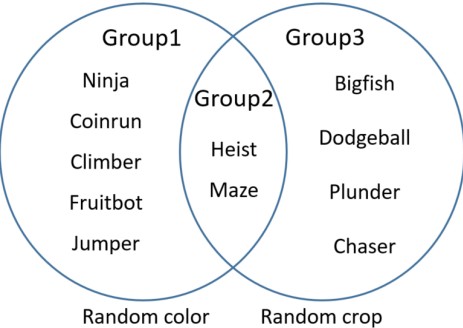

Figure 23: Group of OpenAI Procgen environments: We classify environments according to characteristics about a type of observation window and the importance of color. For Group1, we do not use random crop because they have agent centered window, thus the *random crop* can make hard to find agent. For Group3, we do not use random color, because they have important meaning in the color of objects, thus the transformation of color would make hard to learn information in color. Group2 can apply both of the transformations.

| Augmentation | Task | PPO | RAD | DrAC | DrAC+PAGrad | InDA | ExDA |
|---|---|---|---|---|---|---|---|
| Grayscale | Train | **1.00** | 0.94 | 0.93 | 0.94 | 0.95 | 0.99 |
| | Test-bg | 1.00 | 1.04 | 1.03 | 1.04 | 0.97 | **1.13** |
| | Test-lv | **1.00** | 0.81 | 0.84 | 0.84 | 0.84 | 0.86 |
| Cutout color | Train | **1.00** | 0.72 | 0.82 | 0.84 | 0.76 | 0.94 |
| | Test-bg | 1.00 | 1.33 | 1.27 | 1.29 | 1.19 | **1.53** |
| | Test-lv | **1.00** | 0.69 | 0.83 | 0.83 | 0.69 | 0.93 |

Table 7: Train and test score of InDA and ExDA on Open AI Procgen, compared to baselines PPO, Drac [26], RAD [21], DrAC+PAGrad. **Boldface** indicates the best method.

## K.1 Random convolution

| Easy | PPO | DrAC | Rand_FM | RAD | InDA | ExDA |
|---|---|---|---|---|---|---|
| Climber | **8.63** | 8.33 | 8.27 | 7.93 | 8.5 | 8.1 |
| | ±0.462 | ±0.407 | ±0.187 | ±0.37 | ±0.291 | ±0.268 |
| Jumper | 8.55 | 8.62 | 8.47 | 8.51 | **8.94** | 8.5 |
| | ±0.168 | ±0.075 | ±0.13 | ±0.102 | ±0.09 | ±0.183 |
| Ninja | 7.49 | 8.57 | 7.69 | 7.9 | **8.88** | 7.03 |
| | ±0.421 | ±0.069 | ±0.529 | ±0.652 | ±0.343 | ±0.058 |
| Avg | 1.00 | 1.04 | 0.99 | 0.99 | 1.07 | 0.96 |

Table 8: Training performance benchmark on *easy* with *random convolution*.

| Easy | PPO | DrAC | Rand_FM | RAD | InDA | ExDA |
|---|---|---|---|---|---|---|
| Climber | 6.96 | 7.21 | 6.63 | 6.08 | **7.28** | 7.06 |
| | ±0.651 | ±0.447 | ±0.39 | ±0.264 | ±0.341 | ±0.541 |
| Jumper | 6.85 | **7.97** | 6.7 | 6.74 | 7.94 | 7.54 |
| | ±0.192 | ±0.128 | ±0.167 | ±0.299 | ±0.185 | ±0.158 |
| Ninja | 6.29 | 6.18 | 6.22 | 6.22 | **6.5** | 5.56 |
| | ±0.529 | ±0.193 | ±0.57 | ±0.324 | ±0.191 | ±0.158 |
| Avg | 1.00 | 1.06 | 0.97 | 0.95 | **1.08** | 1 |

Table 9: Test performance benchmark on unseen backgrounds (*easy, random convolution*).

| Easybg | PPO | Oracle | DrAC | Rand-FM | RAD | InDA | ExDA |
|---|---|---|---|---|---|---|---|
| Climber | **12.35** | 9.78 | 11.23 | 12.2 | 12.15 | 10.89 | 12.07 |
| | ±0.083 | ±0.306 | ±0.353 | ±0.128 | ±0.09 | ±0.162 | ±0.073 |
| Coinrun | **9.64** | 7.11 | 9.17 | 9.57 | 9.56 | 8.81 | 9.44 |
| | ±0.07 | ±0.205 | ±0.161 | ±0.126 | ±0.107 | ±0.992 | ±0.149 |
| Fruitbot | 29.78 | 29.74 | 26.07 | **30.19** | 29.92 | 26.17 | 28.76 |
| | ±0.899 | ±0.443 | ±0.658 | ±0.512 | ±0.623 | ±0.575 | ±0.79 |
| Heist | **9** | 7.21 | 5.95 | 7.7 | 7.94 | 5.15 | 8.72 |
| | ±0.513 | ±0.27 | ±0.343 | ±0.6 | ±0.919 | ±0.614 | ±0.533 |
| Jumper | 8.95 | 8.72 | 8.86 | 8.91 | **9.04** | 8.78 | 8.94 |
| | ±0.066 | ±0.119 | ±0.088 | ±0.13 | ±0.135 | ±0.172 | ±0.048 |
| Maze | **9.75** | 8.56 | 8.1 | 9.61 | 9.51 | 9.12 | 9.73 |
| | ±0.513 | ±0.27 | ±0.343 | ±0.6 | ±0.919 | ±0.614 | ±0.533 |
| Ninja | 9.75 | 7.81 | 9.43 | 9.75 | **9.78** | 9.53 | 9.7 |
| | ±0.073 | ±0.422 | ±0.109 | ±0.084 | ±0.03 | ±0.113 | ±0.062 |
| Avg | **1.00** | 0.85 | 0.98 | 0.98 | 0.88 | 0.88 | 0.98 |

Table 10: Training performance benchmark on *easybg* with *random convolution*.

| Easybg | PPO | Oracle | DrAC | Rand-FM | RAD | InDA | ExDA |
|---|---|---|---|---|---|---|---|
| Climber | 1.97 | 9.78 | 7.13 | 2 | 2.34 | 7.36 | **8.11** |
| | ±0.51 | ±0.306 | ±0.419 | ±0.59 | ±1.258 | ±0.273 | ±0.457 |
| Coinrun | 5.48 | 7.11 | 7.54 | 5.65 | 5.48 | 7.14 | **7.81** |
| | ±0.583 | ±0.205 | ±0.188 | ±0.216 | ±0.542 | ±0.479 | ±0.388 |
| Fruitbot | 10.83 | 29.74 | 19.77 | 15.19 | 11.61 | 21.93 | **23.57** |
| | ±1.908 | ±0.443 | ±0.77 | ±3.363 | ±4.615 | ±0.664 | ±0.745 |
| Heist | 5.18 | 7.21 | 5.47 | 5.03 | 4.78 | 4.96 | **8.15** |
| | ±0.838 | ±0.27 | ±0.326 | ±0.6 | ±0.785 | ±0.777 | ±0.633 |
| Jumper | 3.38 | 8.72 | 8.14 | 4.12 | 3.77 | **8.16** | 7.87 |
| | ±0.368 | ±0.119 | ±0.17 | ±0.514 | ±0.435 | ±0.231 | ±0.485 |
| Maze | 6.48 | 8.56 | 6.4 | 6.6 | 6.29 | 8.41 | **8.92** |
| | ±0.523 | ±0.665 | ±0.419 | ±0.494 | ±0.466 | ±0.436 | ±0.155 |
| Ninja | 3.83 | 7.81 | 6.8 | 3.36 | 3.98 | 6.61 | **6.85** |
| | ±0.462 | ±0.422 | ±0.243 | ±0.505 | ±0.44 | ±0.327 | ±0.25 |
| Avg | 1.00 | 2.33 | 1.86 | 1.08 | 1.04 | 1.92 | **2.11** |

Table 11: Test performance benchmark on unseen backgrounds (*easybg, random convolution*).

## K.2 Crop

| Easybg | PPO | DrAC | RAD | InDA | ExDA |
|---|---|---|---|---|---|
| Bigfish | 14.08 ±2.229 | 15.92 ±1.535 | 5.05 ±3.718 | **19.35** ±2.792 | 11.07 ±3.683 |
| Chaser | 5.63 ±0.467 | 3.97 ±0.642 | 1.24 ±0.253 | **6.52** ±0.825 | 4.81 ±0.325 |
| Dodgeball | 7.71 ±0.678 | 10.74 ±0.711 | 1.23 ±0.944 | **12.74** ±1.729 | 6.74 ±0.815 |
| Heist | **9** ±0.513 | 7.58 ±0.11 | 4.53 ±0.266 | 8.15 ±0.57 | 8.79 ±0.424 |
| Maze | **9.75** ±0.033 | 9.03 ±0.348 | 3.95 ±3.418 | 9.63 ±0.143 | 9.72 ±0.026 |
| Plunder | 7.18 ±0.73 | 10.73 ±1 | 0 ±0 | **10.29** ±0.285 | 6.59 ±1.108 |
| Avg | 1.00 | 1.08 | 0.28 | **1.25** | 0.91 |

Table 12: Training performance benchmark on *easybg* with *crop*.

| Easybg | PPO | DrAC | RAD | InDA | ExDA |
|---|---|---|---|---|---|
| Bigfish | 7.43 ±1.65 | 13.63 ±1.504 | 4.93 ±3.696 | **15.19** ±2.724 | 6.35 ±2.466 |
| Chaser | 4.83 ±0.56 | 3.59 ±0.519 | 1.2 ±0.259 | **5.86** ±0.745 | 4.48 ±0.379 |
| Dodgeball | 3.78 ±0.659 | 9.26 ±0.685 | 1.11 ±0.831 | **11.92** ±1.556 | 3.79 ±0.748 |
| Heist | **4.13** ±0.146 | 5.4 ±0.448 | 3.81 ±0.412 | **5.91** ±0.516 | 5.35 ±0.22 |
| Maze | **6.79** ±0.158 | 7.77 ±0.328 | 3.9 ±3.377 | **8.01** ±0.288 | **7.74** ±0.054 |
| Plunder | 5.94 ±0.698 | **9.49** ±0.605 | 0 ±0 | **8.98** ±0.369 | 5.98 ±0.944 |
| Avg | 1.00 | 1.519 | 0.459 | **1.798** | 1.094 |

Table 13: Test performance benchmark on unseen levels (*easybg, crop*).

## K.3 Color jitter

| Easy | PPO | DrAC | RAD | InDA | ExDA |
|---|---|---|---|---|---|
| Climber | 8.5 ±0.575 | **9.33** ±0.212 | 8.64 ±0.156 | 9.43 ±0.21 | 8.18 ±0.45 |
| Jumper | 8.54 ±0.22 | 8.64 ±0.135 | 8.63 ±0.17 | **8.92** ±0.174 | 8.44 ±0.185 |
| Ninja | 7.48 ±0.324 | 8.69 ±0.331 | 8.24 ±0.251 | **9.23** ±0.081 | 7.37 ±0.212 |
| Avg | 1.00 | 1.09 | 1.04 | **1.13** | 0.98 |

Table 14: Training performance benchmark on *easy* with *color jitter*.

| Easy | PPO | DrAC | RAD | InDA | ExDA |
|---|---|---|---|---|---|
| Climber | 6.92 | 8.53 | 8.37 | **8.66** | 8.14 |
| | ±0.761 | ±0.422 | ±0.023 | ±0.24 | ±0.477 |
| Jumper | 6.89 | 7.58 | 7.86 | **7.97** | 7.25 |
| | ±0.223 | ±0.053 | ±0.297 | ±0.292 | ±0.131 |
| Ninja | 6.39 | 6.79 | 7.31 | **7.57** | 6.2 |
| | ±0.585 | ±0.32 | ±0.613 | ±0.555 | ±0.085 |
| Avg | 1.00 | 1.13 | 1.16 | **1.2** | 1.07 |

Table 15: Test performance benchmark on unseen backgrounds (*easy, color jitter*).

| Easybg | PPO | Oracle | DrAC | RAD | InDA | ExDA |
|---|---|---|---|---|---|---|
| Climber | **12.31** | 9.85 | 11.84 | 12 | 11.94 | 12.04 |
| | ±0.092 | ±0.298 | ±0.223 | ±0.256 | ±0.071 | ±0.152 |
| Coinrun | 9.61 | 7.2 | 8.94 | 8.62 | **9.74** | 9.45 |
| | ±0.074 | ±0.195 | ±0.285 | ±0.091 | ±0.05 | ±0.09 |
| Fruitbot | 30.2 | 29.69 | **30.05** | 29.48 | 26.87 | 29 |
| | ±0.691 | ±0.619 | ±0.611 | ±0.507 | ±0.912 | ±0.878 |
| Heist | **8.82** | 7.33 | 7.22 | 6.89 | 7.63 | 8.53 |
| | ±0.523 | ±0.308 | ±0.76 | ±0.348 | ±0.338 | ±0.307 |
| Jumper | 8.97 | 8.67 | 8.9 | 8.94 | **9.03** | **9.03** |
| | ±0.075 | ±0.132 | ±0.05 | ±0.029 | ±0.123 | ±0.086 |
| Maze | **9.75** | 8.08 | 9.46 | 9.46 | 9.3 | 9.67 |
| | ±0.035 | ±0.101 | ±0.404 | ±0.184 | ±0.379 | ±0.111 |
| Ninja | 9.74 | 7.56 | 9.52 | 9.65 | **9.75** | 9.54 |
| | ±0.087 | ±0.286 | ±0.393 | ±0.112 | ±0.046 | ±0.171 |
| Avg | **1.00** | 0.85 | 0.95 | 0.94 | 0.96 | 0.98 |

Table 16: Training performance benchmark on *easbg* with *color jitter*.

| Easybg | PPO | Oracle | DrAC | RAD | InDA | ExDA |
|---|---|---|---|---|---|---|
| Climber | 1.82 | 9.85 | **5.05** | 4.31 | 4.25 | 4.34 |
| | ±0.605 | ±0.298 | ±0.407 | ±0.492 | ±0.338 | ±0.856 |
| Coinrun | 5.42 | 7.2 | 6.46 | 6.47 | **7.13** | 6.53 |
| | ±0.744 | ±0.195 | 0±.526 | ±0.194 | ±0.372 | ±0.375 |
| Fruitbot | 11.78 | 29.69 | 9.49 | 8.51 | 10.88 | **18** |
| | ±1.949 | ±0.619 | ±8.098 | ±1.941 | ±2.263 | ±7.442 |
| Heist | 4.79 | 7.33 | 5.65 | 5.39 | **5.66** | 5.43 |
| | ±0.323 | ±0.308 | ±0.984 | ±0.745 | ±0.271 | ±0.508 |
| Jumper | 3.3 | 8.6 | 5.65 | 5.67 | **5.81** | 5.31 |
| | ±0.467 | ±0.132 | ±0.09 | ±0.953 | ±0.369 | ±0.351 |
| Maze | 6.52 | 8.08 | 8.22 | 8.26 | 8.35 | **8.65** |
| | ±0.304 | ±0.101 | ±0.455 | ±0.175 | ±0.238 | ±0.017 |
| Ninja | 3.56 | 7.56 | 4.22 | 4.18 | **4.34** | 4.07 |
| | ±0.363 | ±0.286 | ±0.487 | ±0.475 | ±0.345 | ±0.332 |
| Avg | 1.00 | 2.4 | 1.44 | 1.37 | 1.43 | **1.48** |

Table 17: Test performance benchmark on unseen backgrounds (*easybg, color jitter*).

| Easybg | PPO | Oracle | DrAC | RAD | InDA | ExDA |
|--------|-----|--------|------|-----|------|------|
| Climber | **12.31** | 9.85 | 11.12 | 11.84 | 11.9 | 12.06 |
| | ±0.092 | ±0.298 | ±0.26 | ±0.505 | ±0.115 | ±0.03 |
| Coinrun | 9.61 | 7.2 | 9.53 | 9.49 | **9.74** | 9.48 |
| | ±0.074 | ±0.195 | ±0.135 | ±0.188 | ±0.046 | ±0.08 |
| Fruitbot | **30.2** | 29.69 | 30.01 | 29.6 | 28.03 | 29.32 |
| | ±0.691 | ±0.619 | ±0.572 | ±0.27 | ±0.994 | ±0.937 |
| Heist | **8.82** | 7.33 | 6.24 | 6.53 | 5.51 | 8.51 |
| | ±0.523 | ±0.308 | ±0.214 | ±0.474 | ±0.146 | ±0.225 |
| Jumper | 8.54 | 8.67 | 8.91 | 8.93 | **9.18** | 8.95 |
| | ±0.22 | ±0.132 | ±0.19 | ±.247 | ±0.18 | ±0.075 |
| Maze | **9.75** | 8.08 | 9.46 | 9.48 | 9.2 | **9.75** |
| | ±0.035 | ±0.101 | ±0.192 | ±0.08 | ±0.367 | ±0.087 |
| Ninja | **9.74** | 7.56 | 9.73 | 9.61 | 9.6 | 9.72 |
| | ±0.087 | ±0.286 | ±0.045 | ±0.096 | ±0.081 | ±0.021 |
| Avg | **1** | 0.85 | 0.93 | 0.94 | 0.95 | 0.99 |

Table 18: Training performance benchmark on *easybg* with *gray*.

## K.4   Gray

| Easybg | PPO | Oracle | RAD | DrAC | InDA | ExDA |
|--------|-----|--------|-----|------|------|------|
| Climber | 1.82 | 9.85 | 1.75 | 1.81 | 1.24 | **2.45** |
| | ±0.605 | ±0.298 | ±0.654 | ±0.211 | ±0.502 | ±0.727 |
| Coinrun | 5.42 | 7.2 | 5.34 | 5.31 | **6.05** | 5.79 |
| | ±0.744 | ±0.195 | ±0.751 | ±0.501 | ±0.465 | ±0.061 |
| Fruitbot | 11.78 | 29.69 | **17.57** | 15.47 | 15.12 | 15.81 |
| | ±1.949 | ±0.619 | ±0.191 | ±1.449 | ±0.958 | ±0.11 |
| Heist | 4.79 | 7.33 | **5.43** | 5.15 | 4.32 | 5.1 |
| | ±0.323 | ±0.308 | ±0.18 | ±0.172 | ±0.112 | ±0.504 |
| Jumper | **6.89** | 8.67 | 2.7 | 4.07 | 3.55 | 4.47 |
| | ±0.223 | ±0.132 | ±0.894 | ±0.46 | ±0.992 | ±0.415 |
| Maze | 6.52 | 8.08 | 7.77 | 7.93 | 7.67 | **8.33** |
| | ±0.304 | ±0.101 | ±0.611 | ±0.104 | ±0.312 | ±0.119 |
| Ninja | 3.56 | 7.56 | 3.72 | 3.91 | 4.02 | **4.03** |
| | ±0.363 | ±0.286 | ±0.131 | ±0.62 | ±0.666 | ±0.071 |
| Avg | 1 | 2.2 | 1.03 | 1.04 | 0.97 | **1.13** |

Table 19: Test performance benchmark on unseen backgrounds (*easybg, gray*).

| Easy | PPO | DrAC | RAD | InDA | ExDA |
|------|-----|------|-----|------|------|
| Climber | **8.5** | 6.95 | 7.55 | 7.22 | 8.05 |
| | ±0.575 | ±0.547 | ±0.256 | ±0.312 | ±0.461 |
| Jumper | 8.54 | 8.4 | 8.58 | **8.85** | 8.5 |
| | ±0.22 | ±0.224 | ±0.199 | ±0.015 | ±0.224 |
| Ninja | 7.48 | 6.67 | 7.1 | **8.91** | 7.05 |
| | ±0.324 | ±0.435 | ±0.718 | ±0.165 | ±0.24 |
| Avg | 1 | 0.9 | 0.95 | **1.026** | 0.96 |

Table 20: Training performance benchmark on *easybg* with *gray*.

| Easy | PPO | DrAC | RAD | InDA | ExDA |
|---|---|---|---|---|---|
| Climber | 6.92 | 4.49 | 5.57 | 5.11 | **7.24** |
| | ±0.761 | ±0.332 | ±0.307 | ±0.483 | ±0.721 |
| Jumper | **6.89** | 5.38 | 6.59 | 6.35 | 6.87 |
| | ±0.223 | ±0.215 | ±0.055 | ±0.234 | ±0.182 |
| Ninja | 6.39 | 5.67 | 5.14 | **6.84** | 6.01 |
| | ±0.585 | ±0.318 | ±0.628 | ±0.206 | ±0.651 |
| Avg | **1** | 0.77 | 0.86 | 0.91 | 0.99 |

Table 21: Test performance benchmark on unseen backgrounds (*easy, gray*).

## K.5 Cutout color

| Easybg | PPO | Oracle | DrAC | RAD | InDA | ExDA |
|---|---|---|---|---|---|---|
| Climber | **12.31** | 9.85 | 11.92 | 8.26 | 11.76 | 12.07 |
| | ±0.092 | ±0.298 | ±0.158 | ±0.663 | ±0.027 | ±0.127 |
| Coinrun | 9.61 | 7.2 | 9.23 | 8.07 | **9.7** | 9.39 |
| | ±0.074 | ±0.195 | ±0.323 | ±0.645 | ±0.084 | ±0.012 |
| Fruitbot | **30.2** | 29.69 | 29.73 | 29.2 | 27.18 | 28.95 |
| | ±0.691 | ±0.619 | ±0.898 | ±0.64 | ±1.302 | ±0.907 |
| Heist | **8.82** | 7.33 | 8.47 | 6.25 | 6.1 | 8.65 |
| | ±0.523 | ±0.308 | ±0.397 | ±0.704 | ±0.693 | ±0.21 |
| Jumper | 8.97 | 8.67 | 8.87 | 8.75 | **9.1** | 8.91 |
| | ±0.075 | ±0.132 | ±0.123 | ±0.131 | ±0.081 | ±0.053 |
| Maze | **9.75** | 8.08 | 9.41 | 9.17 | 9.27 | 9.74 |
| | ±0.035 | ±0.101 | ±0.134 | ±0.118 | ±0.125 | ±0.133 |
| Ninja | **9.74** | 7.56 | 9.65 | 7.17 | 9.72 | 9.7 |
| | ±0.087 | ±0.286 | ±0.138 | ±1.993 | ±0.02 | ±0.02 |
| Bigfish | **13.89** | 13.22 | 2.54 | 5.19 | 1.95 | 11.22 |
| | ±3.127 | ±1.488 | ±0.13 | ±3.658 | ±0.311 | ±3.66 |
| Chaser | **5.49** | 3.04 | 2.88 | 1.98 | 3.34 | 5 |
| | ±0.562 | ±0.183 | ±0.699 | ±0.112 | ±0.755 | ±0.187 |
| Dodgeball | **7.76** | 5.74 | 5.71 | 5.98 | 2.79 | 6.57 |
| | ±0.859 | ±1.118 | ±1.008 | ±0.103 | ±1.612 | ±0.693 |
| Plunder | **7.15** | 6.05 | 5.43 | 4.34 | 4.92 | 6.87 |
| | ±0.95 | ±0.58 | ±0.082 | ±0.24 | ±0.625 | ±1.255 |
| Avg | **1.00** | 0.82 | 0.82 | 0.72 | 0.76 | 0.94 |

Table 22: Training performance benchmark on *easybg* with *cutout color*.

| Easy | PPO | Oracle | DrAC | RAD | InDA | ExDA |
|---|---|---|---|---|---|---|
| Climber | 8.5 | 9.85 | 7.69 | 6.67 | **9.02** | 8.02 |
| | ±0.575 | ±0.298 | ±0.237 | ±0.381 | ±0.473 | ±0.506 |
| Jumper | 7.48 | 7.56 | 6.28 | 5.6 | **8.57** | 7.41 |
| | ±0.324 | ±0.286 | ±0.257 | ±0.276 | ±0.122 | ±0.125 |
| Ninja | 8.54 | 8.67 | 8.45 | 8.32 | **8.93** | 8.53 |
| | ±0.22 | ±0.132 | ±0.183 | ±0.051 | ±0.166 | ±0.095 |
| Avg | 1.00 | 1.06 | 0.91 | 0.84 | **1.08** | 0.98 |

Table 23: Training performance benchmark on *easy* with *cutout color*.

| Easybg | PPO | Oracle | DrAC | RAD | InDA | ExDA |
|---|---|---|---|---|---|---|
| Climber | 1.82 | 9.85 | 3.54 | 3.97 | 3.4 | **4.29** |
| | ±0.605 | ±0.298 | ±0.164 | ±0.999 | ±0.645 | ±0.154 |
| Coinrun | 5.42 | 7.2 | 5.87 | 5.93 | 6.2 | **6.41** |
| | ±0.744 | ±0.195 | ±.251 | ±0.061 | ±0.357 | ±0.131 |
| Fruitbot | 11.78 | 29.69 | 18.18 | **19.24** | 17.69 | 17.7 |
| | ±1.949 | ±0.619 | ±3.744 | ±3.385 | ±4.026 | ±0.888 |
| Heist | 4.79 | 7.33 | 6.6 | 5.76 | 4.97 | **7.51** |
| | ±0.323 | ±0.308 | ±.092 | ±0.551 | ±0.33 | ±0.119 |
| Jumper | 3.3 | 8.67 | 4.99 | 5.48 | 5.43 | **6.02** |
| | ±0.467 | ±0.132 | ±0.114 | ±0.28 | ±1.116 | ±0.235 |
| Maze | 6.52 | 8.08 | 7.33 | 7.66 | 7.01 | **7.83** |
| | ±0.304 | ±0.101 | ±0.223 | ±.243 | ±0.17 | ±0.22 |
| Ninja | 3.56 | 7.56 | **4.29** | 3.96 | 3.75 | 3.76 |
| | ±0.363 | ±0.286 | ±0.245 | ±0.152 | ±0.333 | ±0.348 |
| Bigfish | 3.4 | 13.22 | 1.29 | 2.5 | 1.29 | **4.49** |
| | ±0.487 | ±1.488 | ±0.08 | ±2.331 | ±0.152 | ±0.776 |
| Chaser | 0.91 | 3.04 | 1.08 | 1.13 | 1.68 | **1.73** |
| | ±0.061 | ±0.183 | ±0.038 | ±0.157 | ±0.305 | ±0.698 |
| Dodgeball | 2.17 | 5.74 | 3.92 | 4.02 | 1.97 | **4.37** |
| | ±0.53 | ±1.118 | ±0.53 | ±0.345 | ±1.098 | ±0.527 |
| Plunder | **6.87** | 6.05 | 5.27 | 4.77 | 4.71 | 6.45 |
| | ±0.933 | ±0.58 | ±0.208 | ±0.612 | ±0.622 | ±1.232 |
| Avg | 1.00 | 2.51 | 1.27 | 1.33 | 1.19 | **1.53** |

Table 24: Test performance benchmark on unseen backgrounds (*easybg, cutout color*).

| Easy | PPO | Oracle | DrAC | RAD | InDA | ExDA |
|---|---|---|---|---|---|---|
| Climber | 6.92 | 9.85 | 6.54 | 5.24 | **7.61** | 7.25 |
| | ±0.761 | ±0.298 | ±0.213 | ±0.417 | ±0.486 | ±0.325 |
| Jumper | 6.39 | 7.56 | 5.06 | 4.9 | **6.71** | 5.78 |
| | ±0.585 | ±0.286 | ±0.137 | ±0.382 | ±0.352 | ±0.488 |
| Ninja | 6.89 | 8.67 | 6.88 | 6.79 | 6.81 | **6.92** |
| | ±0.223 | ±0.132 | ±0.083 | ±0.278 | ±0.355 | ±0.212 |
| Avg | 1.00 | 1.29 | 0.91 | 0.84 | **1.05** | 0.99 |

Table 25: Test performance benchmark on unseen backgrounds (*easy, cutout color*).

| Easybg | PPO | DrAC | RAD | InDA | ExDA |
|--------|-----|------|-----|------|------|
| Climber | **11.14** | 10.77 | 7.26 | 9.45 | 10.75 |
|  | ±0.077 | ±0.279 | ±0.843 | ±0.193 | ±0.114 |
| Coinrun | **8.64** | 8.36 | 6.89 | 7.76 | 8.32 |
|  | ±0.05 | ±0.348 | ±0.503 | ±0.096 | ±0.224 |
| Fruitbot | **28.26** | 26.88 | 26.22 | 23.79 | 26.33 |
|  | ±0.461 | ±1.276 | ±1.258 | ±0.971 | ±0.894 |
| Heist | **4.07** | 3.92 | 2.27 | 2.15 | 3.93 |
|  | ±0.07 | ±0.276 | ±0.448 | ±0.553 | ±0.184 |
| Jumper | **7.38** | 7.32 | 6.98 | 6.68 | 7.25 |
|  | ±0.15 | ±0.195 | ±0.199 | ±0.24 | ±0.117 |
| Maze | 6.8 | **6.84** | 6.04 | 5.91 | 6.17 |
|  | ±0.2 | ±0.137 | ±0.258 | ±0.03 | ±0.162 |
| Ninja | 8.56 | **8.63** | 6.28 | 7.81 | 8.34 |
|  | ±0.061 | ±0.132 | ±1.866 | ±0.21 | ±0.119 |
| Bigfish | **7.16** | 0.91 | 2.29 | 0.95 | 6.04 |
|  | ±2.263 | ±0.037 | ±2.306 | ±0.06 | ±2.783 |
| Chaser | **4.54** | 2.61 | 1.8 | 2.47 | 4.22 |
|  | ±0.503 | ±0.509 | ±0.12 | ±0.506 | ±0.331 |
| Dodgeball | **3.78** | 2.71 | 2.53 | 1.26 | 2.82 |
|  | ±0.823 | ±0.362 | ±0.135 | ±0.48 | ±0.593 |
| Plunder | **5.99** | 5.08 | 4.07 | 4.55 | 5.83 |
|  | ±0.814 | ±0.305 | ±0.479 | ±0.393 | ±1.061 |
| Avg | **1.00** | 0.83 | 0.69 | 0.69 | 0.93 |

Table 26: Test performance benchmark on unseen levels (*easybg, cutout color*).

| Easy | PPO | DrAC | RAD | InDA | ExDA |
|------|-----|------|-----|------|------|
| Climber | 5.45 | **5.9** | 5.3 | 4.26 | 5.71 |
|  | ±0.77 | ±0.352 | ±0.307 | ±0.122 | ±0.303 |
| Jumper | 5.81 | **6.01** | 4.93 | 4.56 | 5.43 |
|  | ±0.227 | ±0.389 | ±0.08 | ±0.161 | ±0.333 |
| Ninja | 5.77 | 5.67 | 5.8 | 5.65 | **5.87** |
|  | ±0.09 | ±0.023 | ±0.071 | ±0.166 | ±0.079 |
| Avg | 1.00 | **1.03** | 0.94 | 0.85 | 1 |

Table 27: Test performance benchmark on unseen levels (*easy, cutout color*).

## L   Primitive evaluation on DeepMind Control Suite with SAC

We experiment on DeepMind Control Suite (DMC) with a preliminary experiment which provides justification for our proposed method ExDA due to the limited time and computation resources. We first note that [26] has demonstrated that UCB-DrAC, which is similar to UCB-InDA, can accelerate RL training in DMC. Hence, in order to justify our proposed method UCB-ExDA, which combines UCB-InDA and ExDA, it would be sufficient to show the existence of the cases showing a benefit of ExDA in DMC. Furthermore, we use SAC as the base RL algorithm instead of PPO to show the versatility of our method ExDA. SAC is off-policy actor-critic RL algorithm which trains actor and critic networks separately with a replay buffer. Thus, we can use stored data in the replay buffer after RL training, and also we can only distill the output of the actor-network, because the actor-network determines the policy $\pi$ from observations, while the critic-network computes the value function $V$ from observations. We denote the actor-network parameters using $\theta$ same as the notation of PPO in Section 4. We ablate the effect of ExDA to compare w/ and w/o inconsistency loss of policy on original observations. Thus ExDA is trained with Eq 5 after SAC, but ablated ExDA is trained with Eq 11.

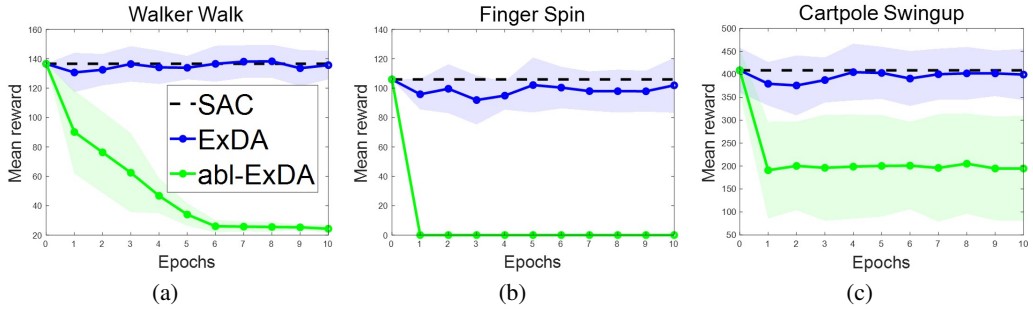

Figure 24: Ablation of ExDA on DMcontrol

$$L_{abl-\mathrm{ExDA}}(\theta, \phi; \theta_{\mathrm{old}}) := L_{\mathrm{dis}}(\theta, \phi; \theta_{\mathrm{old}}) . \tag{11}$$

Both methods are trained 10 epochs after 10K time-steps SAC training and detailed hyper-parameters are described below the table. We compare the mean reward of each method SAC, ExDA, abl-ExDA on three environments *Walker-Walk, Finger-Spin, Cartpole-Swingup* and 5 seeds. We evaluate each reward from 50 evaluation episodes in train environments. As below Figure 24, ExDA almost maintains the mean reward of SAC, abl-ExDA degrades the performance of SAC by distilling policy only to augmented observations except the original.

| Hyperparameter | Value |
|---|---|
| # of action repeat | 8 |
| # of frame stack | 3 |
| Data augmentation | Random Convolution |
| Learning rate of DA | $10^{-2}$ |
| Batch size | 128 |
| # of train steps | 100000 |
| # of distill epochs | 10 |
| Replay buffer capacity | 100000 |
| Init steps | 1000 |
| Learning rate of critic | $10^{-3}$ |
| $\beta$ of critic | 0.9 |
| $\tau$ of critic | 0.01 |
| Target update frequency of critic | 2 |
| Learning rate of actor | $10^{-3}$ |
| $\beta$ of actor | 0.9 |
| Log std min of actor | -10 |
| Log std max of actor | 2 |
| Update frequency of actor | 2 |
| Encoder type | pixel |
| Feature dimension of encoder | 50 |
| Learning rate of encoder | $10^{-3}$ |
| $\tau$ of encoder | 0.05 |
| # of layers | 4 |
| # of filters | 32 |
| Latent dimension | 128 |
| Discount factor | 0.99 |
| Learning rate of $\alpha$ | $10^{-4}$ |
| $\beta$ of $\alpha$ | 0.5 |