# OpenReview forum: "Efficient Scheduling of Data Augmentation for Deep Reinforcement Learning"
_NeurIPS.cc/2022/Conference — NeurIPS 2022 Accept_

### Official Review · Reviewer_SwS3 · 2022-06-29

**Rating:** 6
**Confidence:** 4
**Soundness:** 2 fair
**Presentation:** 2 fair
**Contribution:** 2 fair

**Summary:**

This paper explores several modifications to UCB-DrAC [25]. It decouples the optimization of the RL loss and the data-augmentation regularization losses in two separate stages, and then empirically studies what is the best strategy for scheduling those two stages: alternating between them at fixed intervals during training (InDA), or training with RL only followed by a few epochs of regularization (ExDA). It also augments the UCB bandit in UCB-DrAC with an arm that does not perform any type of data augmentation. These methods are evaluated on a variant of the Procgen benchmark, where they compare favorably to the considered baselines (PPO, RAD, DrAC and DrAC+PAGrad).

**Questions:**

**Major**

I am particularly concerned about the following points described under *weaknesses* above which I believe should be resolved before this paper is ready for publication at NeurIPS:
- Results under standard Procgen evaluation protocol
- Ensuring that baselines replicate published results
- Better positioning of the proposed method with respect to the literature

**Minor**

- InDA and ExDA require additional learner steps compared to methods that optimize both the RL and the consistency loss in parallel (e.g. DrAC). What's the impact on wall-clock time?
- It would be interesting to understand how the performance of InDA and ExDA changes as a function of the size of the replay buffer used for distillation. In particular, I would expect a collapse if this buffer is too small. Have authors experimented with this?
- Figure 9 (Appendix D) shows train scores for different number of backgrounds. It is somewhat expected that increasing the number of backgrounds will make training slower, given that the agent is exposed to a more difficult task distribution. It would be interesting to compare those same experiments in the test set, where agents trained on fewer background might underperform due to overfitting.

**Limitations:**

Section 6 briefly mentions one limitation of the approach (i.e. the framework is specialized, and thus restricted, to RL). The paper does not describe potential negative societal impacts, although there aren't any specific to the proposed approach.

**Strengths And Weaknesses:**

**Strengths**

- Generalization in RL is an important topic and the paper provides interesting insights about how data augmentation interacts with RL training.
- The proposed methods are simple to implement on top of DrAC and could be easily adopted by practitioners.


**Weaknesses**

- The evaluation protocol differs from [25] despite the proposed methods are a direct extension of DrAC. If I understood it correctly, authors consider a more challenging case where the agents are only trained on a single background per task (*easybg*) instead of Procgen's *easy* setup; this is motivated in L203 as a way to *see clear advantage from visual augmentations*. Given that [25] already showed an advantage of using augmentations in the standard Procgen benchmark, this choice is difficult to understand. I would strongly encourage including results on both the original setting as well.
- While the difference in evaluation protocol makes is difficult to compare results in this paper with those in the literature, it is surprising to see that DrAC underperforms PPO in most experiments (e.g. Figure 3, Table 2). This contradicts the results of [25]. Why is this the case?
- Given that the methods are an extension of DrAC (splitting the optimization of the RL and consistency losses in two stages, and adding an identity arm to the bandit), presenting DA/InDA/ExDA as entirely novel contributions (e.g. Sections 1 and 4) is somewhat of an overclaim. Note that properly positioning itself as an analysis or extension of DrAC would not reduce the merits of the paper but would make it easier for readers to understand the problems being studied.

---

> ### Author Response · Authors · 2022-08-02
> **Response to reviewer SwS3**
>
> We appreciate reviewer SwS3 for the insightful comments.
>
> **Weakness1**: Justification of experiment on easy-bg environment \
> & **Weakness2**: Understanding lower scores of DrAC (and InDA) than that of PPO in some experiments (e.g., Figure 3 and Table 2) \
> We use easy-bg mode to investigate the case that using data augmentation that is helpful for testing but not for training, i.e., interfering with RL training, and thus ExDA is better than DrAC or InDA, e.g., the first row of Table 2. We note that even in [25] using easy mode, there are several cases showing train score degeneration from using DrAC (Figure 8 of [25]). Furthermore, we also refer reviewer SwS3 to Appendix I of the supplementary material for the experiments when agents are trained on easy mode (the original setup of [25]), which align with our main findings. Finally, we want to note that studying the interference from data augmentation is important, since the consistency with respect to a given data augmentation can be (partially) inappropriate in practice. When only wrong augmentations are given, DrAC obviously fails, while UCB-ExDA can perform fairly.
>
> **Weakness3**: Our contribution of DA/InDA/ExDA compared to DrAC \
> In the revision, we will thoroughly review our manuscript so that we avoid to overclaim our contribution compared to DrAC [25]. As commented, we agree that our major contributions are investigating and alleviating the interference from augmentation.
>
> **Question1**: Comparision of InDA and ExDA to DrAC in terms of computation time \
> The computational complexities of DrAC and InDA are similar since they require similar numbers of forward and backward propagations of original and augmented inputs, in spite of the difference in loss functions. However, ExDA is faster than DrAC and InDA since its RL training requires no computation of distillation loss, i.e., no forward/backward propagations of augmented input, and the distillation requires only few (9) epochs with 0.5M timesteps. We present the above discussion about computational complexity in Appendix F.3.
>
> **Question2**:  Ablation about replay buffer size \
> When we chose the hyperparameters of InDA and ExDA, we experimented with varying buffer size for distillation. As expected, we observed that decreasing buffer size degrades the effect (benefit or interference) from the augmentation. As we propose several ways to mitigate the interference, we make sure that the buffer size is sufficient for a certain level of convergence of inconsistency loss, while we sought a minimal value of such buffer sizes in order to reduce computational complexity. We will add the above discussion in the revision.
>
> **Question3**: Additional experiments related to Figure 9  \
> As expected, training with a few backgrounds is prone to cause overfitting issues. However, if we know/believe that an augmentation regulates the overfitting to the train environment, it can be better to first have the overfitted agent with a high train score (but low test score) and then fix it with ExDA to improve the test score, c.f. the experiment with varying the number $B$ of backgrounds in training. <https://drive.google.com/file/d/1mSLmYEsDwSyw92yVJ2jW4ymBQ8OAYS5W/view?usp=sharing>

---

> > ### Comment · Reviewer_SwS3 · 2022-08-09
> > **Response to authors**
> >
> > Authors have addressed my main concerns, so I am updating my rating to 6.

---

> > > ### Author Response · Authors · 2022-08-09
> > > **Response to reviewer SwS3**
> > >
> > > Thank you for your response, and also we really appreciate for updating your rating.
> > >
> > > We add explanation of easy-bg mode in section 5.1 about Weakness 1.
> > > And you can check any other change with blue text in the paper.

---

### Official Review · Reviewer_fohF · 2022-07-10

**Rating:** 6
**Confidence:** 3
**Soundness:** 3 good
**Presentation:** 3 good
**Contribution:** 2 fair

**Summary:**

This paper proposes two data-augmentation techniques to improve sample efficiency and generalization capability based on timing. (1) intra distillation separates proximal policy optimization and distillation optimization, and is applied during the whole training period; (2) extra distillation is applied after training.

**Questions:**

- From Table 1, it seems that the improvement is not significant.

**Limitations:**

Yes

**Strengths And Weaknesses:**

## Strengths

- It is well-written and easy to understand. The figures describe the proposed paper in an easy-to-understand graphical format.
- the proposed method is simple and effective.

## Weaknesses

### The formulation is not general enough

In the background section, the authors fix the observation representation into two-dimensional square image. It seems to be the implementation detail instead of the formulation. For example, the observation could not be limited to images, the image size could not be limited to be square (i.e., $\mathbb{R}^{m\times n}$).

### Ambiguous statement and lack proof.

In line 146-147, the authours state that the proposed method is more stable than the alternatives. I wonder stability competes in terms of learning theory, sample efficiency, or optimization? Besides the intuitive illustration in Figure 2, can you provide the proof to support your statement?

### Data distribution modelling vs. proposed separation training between $\mathcal{L}_{PPO}$ and $\mathcal{L}_{DA}$]

The authors propose to alternate PPO and DA optimization, what is the benefits compared with directly modeling the data distribution, or can you provide some insights for distribution modeling?

### Lack of ellobration for ExDA

The proposed ExDA is one of the main contributions in this paper, but the authors leave some details of computational cost reduction into the supplementary part. It would be better if you can give some definitions/remarks/proofs to show why removing the value inconsistency measurement helps? If due to the space issue, I suggest combining Figure 3 and Figure 4 to fill the blank area.

### Lack of SOTA baselines
The comparisons between the proposed method and baselines are not thorough enough. For example, CURL[1] and DrQ[2] are missed.

Some minor typos are listed below:
* Line 72, learn with -> to learn with;
* Line  74-75,  a limited applicability -> limited applicability;
* Line 78, a curriculum learning -> curriculum learning;
* Line 203, clear advantage -> a clear advantage.

*Reference*
[1] Laskin, Michael, Aravind Srinivas, and Pieter Abbeel. "Curl: Contrastive unsupervised representations for reinforcement learning." International Conference on Machine Learning. PMLR, 2020.
[2] Yarats, Denis, Ilya Kostrikov, and Rob Fergus. "Image augmentation is all you need: Regularizing deep reinforcement learning from pixels." International Conference on Learning Representations. 2020.

---

> ### Author Response · Authors · 2022-08-02
> **Response to reviewer fohF**
>
> We appreciate reviewer fohF for the detail comments about presentation.
>
> **Weakness1**: More generic description of model \
> We thank reviewer fohF for the comment. As commented, our methods are also applicable to the case with more diverse formats of input than the squared image. We will revise the description of the model to be more generic than the submitted one.
>
> **Weakness2**: Further justification of the benefit from separating RL training and distillation \
> We apologize for the ambiguous statement in L146-147, which we will tone down in the revision. We wanted to point out that the alternatives to distill augmentation are prone to interfere with RL training. Indeed, besides the illustration of Figure 2, we provide an empirical evidence (Figure 3: both train and test scores drop sharply right after applying concurrent optimization of RL and DA losses) showing the importance of explicitly separating RL training and distillation and maintaining the policy from RL training during distillation.
>
> **Weakness3**: Comparison of the proposed DA to data distribution modeling \
> To be honest, we need further clarification of the question. It would be helpful to provide some references about the data distribution modeling. If you wanted to ask the rationale of alternating the optimizations of $L_{\text{PPO}}$ and $L_{\text{DA}}$, then we refer the reviewer to Section 4.1 with the empirical justification of Figure 3 or our response to W2 in the above. If you meant, by data distribution modeling, domain adaptation techniques based on distribution matching (e.g., [Wang et al. “TENT: …” ICLR 2021: <https://openreview.net/pdf?id=uXl3bZLkr3c>]), it is an interesting direction for future work, while it can be limited to a specific class of neural network architectures with some interpretation of feature space.
>
> **Weakness4**: Better presentation of details about ExDA \
> We thank reviewer fohF for the kind suggestion. In the revision, we will reorganize the manuscript so that we can mention the advantage of ExDA in terms of computational complexity in the main text, which was originally postponed to the supplementary material as our main focus is the sample efficiency and generalization performance. \
> As suggested, in the revision, we will also add additional experiment justifying the exclusion of value inconsistency loss from $L_{\text{DA}}$. In fact, this is because the value consistency is not necessary for RL training or distillation in the actor-critic framework, and ensuring it has a potential risk of generating additional interference. Indeed, we empirically observed that including the value inconsistency not only interferes more but also requires extra efforts to find a hyperparameter stabilizing training.
>
> **Weakness5**: Other baselines, e.g., CuRL, DrQ \
> In the final version, we will try our best to include an additional experiment with the suggested baselines, which we are working on. However, we believe our main findings would remain the same with the other baselines: CuRL and DrQ.
>
> **Question1**: The scale of performance gaps in Table 1 \
> The gap can seem small, but it mainly because all the scores are normalized and averaged over a set of games (without any tuning of hyperparameter per game). We also want to note that a large number of additional samples may require to fulfill a small score gap, since the learning curves are typically concave (and increasing).

---

> > ### Comment · Reviewer_fohF · 2022-08-09
> > **Response to rebuttal**
> >
> > I thank the authors for the response and am satisfied with it. Therefore, I decide to change the rating to 6.

---

> > > ### Author Response · Authors · 2022-08-09
> > > **Response to reviewer fohF**
> > >
> > > We appreciate for your response and increasing rating.
> > >
> > > We revise our paper with your advice.
> > > You can check revised part with color blue in the main paper and appendix.

---

### Official Review · Reviewer_ZudK · 2022-07-21

**Rating:** 7
**Confidence:** 3
**Soundness:** 4 excellent
**Presentation:** 3 good
**Contribution:** 3 good

**Summary:**

Data augmentation is a critical regularization tool in supervised learning, used to induce prior domain knowledge into the model. In reinforcement learning however, data augmentation hurts and causes interference with training. In this work, the authors propose simple methods to incorporate data augmentation during learning, improving model generalization while alleviating the interference with training. Additionally, the authors propose a method that adaptively select the appropriate set of augmentations for an RL task.

**Questions:**

1. While InDA and ExDA individually make sense, what would happen if both methods are applying (performing distillation during and after training)?
2. When performing adaptive selection, how do the UCB-ExDA and UCB-InDA methods differ in their choice of the most beneficial augmentations? Do they always converge to the same augmentation?

**Limitations:**

The authors address and discuss the main limitations of the work

**Strengths And Weaknesses:**

Strengths:
1. The authors present a simple and novel framework for using data augmentation in reinforcement learning. The motivation for the method is grounded in observations of the effects of data augmentation in supervised learning. The proposed method addresses the problem of RL training interference in a simple way. The results on the OpenAI Procgen benchmark are convincing and the authors both validate their methods and propose different insights into how augmentation works in the context of RL.
2. This work is relevant and I believe it can of interest to the community at large. Data augmentation has long been a critical component of supervised learning (and self-supervised learning), and it is important to have work such as this that utilized this framework in RL.
3. The authors clearly discuss the differences between the two proposed methods InDA and ExDA, and provide sufficient details to enable the reader to choose between the two. ExDA, in particular, is promising due to its low computational overhead.
4. UCB variants of the two methods are also proposed. They enable the model to find, in a set of augmentations, the most beneficial ones. While this approach is not novel, the addition of the "identity" augmentation appears to be a critical and relevant component to avoid interference with training in particular cases.

Weaknesses:
1. PPO is used as the baseline RL algorithm, no other algorithm is tested, so it is uncertain how general the method and the results are. The same thing goes for the benchmark RL tasks that were considered: all video games, with 16x16 images.
2. In section 5.2, ExDA is applied after 20M timesteps of RL training. After the 20M timesteps, no further RL training is performed, which is not the case for InDA. InDA continues to switch between RL optimization steps and distillation steps. For fair comparison between ExDA and InDA, it might be necessary to have the same number of total RL training steps, i.e. training for 25M steps before introducing ExDA

---

> ### Author Response · Authors · 2022-08-02
> **Response to reviewer ZudK**
>
> We appreciate reviewer ZudK for the thorough review and constructive comments.
>
> **Weakness1**: More experiments with other RL baseline algorithms and environments \
> We are working on an additional experiment on DMControl, which we will definitely include in the final version. In our primitive investigation, our main findings in the submission seem to remain the same. We also want to note that the experiment in the submission has been conducted with an extensive set of ProcGen environments, including 11 games of two modes.
>
> **Weakness2**: Comparison between InDA and ExDA using the same steps of RL training \
> In the submission, we set the evaluation setup to be somewhat unfavorable to ExDA (using 20M steps of RL training followed by additional 0.5M steps of distillation; denoted by ExDA(20M)) compared to InDA or other baselines (25M steps of RL training) to clearly avoid potential complaint about the extra 0.5M steps for ExDA. It is obvious that the performance of ExDA is improved if we put more steps for RL training, and thus the benefit of ExDA compared to InDA becomes more conspicuous if ExDA is the effective timing. Indeed, Heist+rand-color tested with diverse backgrounds is the case as the following result of our additional experiment evaluating ExDA(25M) using 25M RL steps and 0.5M distillation steps:
>
> | **Heist**   | **PPO** | **DrAC** | **RAD** | **InDA** | **ExDA(20M)** | **ExDA(25M)** |
> |-------------|---------|----------|---------|----------|---------------|---------------|
> | **Train**   | **9**   | 5.95     | 7.94    | 5.15     | 8.72          | 8.93          |
> | **Test-bg** | 5.18    | 5.47     | 4.78    | 4.96     | 8.15          | **8.26**      |
> | **Test-lv** | 4.13    | 5.4      | 3.81    | **5.91** | 5.35          | 5.41          |
>
> We thank reviewer ZudK for the suggestion, since the increased gap between InDA and ExDA on Heist(test-bg) reinforces our main message: postponing data augmentation when it generates a severe interference with RL training. ExDA(25M) has a slight drop of train score compared to PPO, but it could be eventually removed if we put (slightly) more steps for distillation. In the revision, we will include the above discussion and result.
>
> **Question1**: Applying both InDA and ExDA  \
> Thanks to the explicit maintenance of RL policy in ExDA (c.f., Figure 2), ExDA followed by InDA for the same augmentation would preserve train and test scores of InDA if InDA has been successfully distilling the augmentation. Hence, in this case, ExDA is not necessary. However, when the distillation of InDA is incomplete, ExDA can be helpful to improve the test score (while there is not much hope to improve train score). To enjoy such a benefit from ExDA, UCB-ExDA (UCB-InDA followed by ExDA) employs ExDA for distilling augmentations which have not selected enough in UCB-InDA.
>
> **Question2**: Difference between UCB-ExDA and UCB-InDA \
> UCB-ExDA is nothing but UCB-InDA followed by ExDA. Therefore, UCB-InDA and UCB-ExDA share the same procedure of identifying and applying the most beneficial augmentation for InDA. However, only UCB-ExDA is able to enjoy further benefit from augmentations that have been distilled incompletely in UCB-InDA (or InDA) at a small cost of extra distillations.

---

> > ### Comment · Reviewer_ZudK · 2022-08-08
> > **Response to rebuttal**
> >
> > I thank the authors for their response, and for providing further clarifications. I remain in favor of accepting the paper.

---

> > > ### Author Response · Authors · 2022-08-09
> > > **Response to reviewer ZudK**
> > >
> > > We thank you for your response.
> > >
> > > We revise the paper by reflecting the suggestions in your review.
> > > We experiment on DeepMind Control Suite with SAC, which is added to Appendix L.
> > > Furthermore, we add study of ExDA with further RL training in Appendix H.
> > >
> > > We really appreciate for your helpful advices.

---

### Official Review · Reviewer_tBmt · 2022-07-23

**Rating:** 7
**Confidence:** 3
**Soundness:** 3 good
**Presentation:** 3 good
**Contribution:** 3 good

**Summary:**

In this paper, the authors investigate the question of how and when to use data augmentation (DA) in the deep reinforcement learning (DRL) training pipeline since it can interfere with the training and its effect is going to be time-sensitive for non-stationary RL. In this context, they propose two DA schemes depending on the use case, whether for sample efficiency (InDA) or for generalization (ExDA). They also propose an adaptive DA scheme, UCB-ExDA, with all the augmentations, including the option to not augment as arms of the MAB problem, followed by ExDA. The idea behind these DA schemes is to have a standalone distillation method at any point of the training-- during it as in the case of InDA, or after it as in the case of ExDA. DRL training and DA distillation are kept separate, and the authors also who the usefulness of postponing DA to the end of DRL training.

**Questions:**

- It would be interesting to see more discussion on DrAC+PAGrad, especially looking at its performance when considering separating out distillation from RL training. The authors say that they devise it as another mechanism to relieve the interference between RL training and augmentation, but why?
- It's interesting that removing the value inconsistency measure from the distillation loss doesn't degrade RL performance-- why would this be so?

**Limitations:**

The authors note that their proposed framework and findings are limited to the unique setting of non-stationary RL. The authors also state that there is potentially no negative societal impact of their work, and to the best of my understanding and imagination, I can also not think of one, since the paper is more investigative in nature.

**Strengths And Weaknesses:**

*Strengths*

The proposed DA strategies are well-motivated and well-discussed and the authors also find significant contrasting time-sensitivities of DA methods, which is interesting and demands further investigation. Distillation losses are well-illustrated and well-explained. Experiments and ablations are helpful-- the timing, performance, and different types of transformations are extensively studied.

*Weaknesses*

- The experimental design is sound, but it would be good to see some experiments on less computationally intensive benchmarks that do not require training for 25mn timesteps-- are there any, why not? Or at least, this should be addressed in the paper.
- Minor/writing: Possible typo on line 189 on Page 5-- N_k(s) is the number of *times*?

---

> ### Author Response · Authors · 2022-08-02
> **Response to reviewer tBmt**
>
> We appreciate reviewer tBmt for the positive feedback and interesting suggestions.
>
> **Weakness1**: Lighter benchmarks for evaluation \
> We focused on OpenAI ProcGen environment since it provides an intensive set of configurations to evaluate different pairs of generalization and augmentation (level&crop; and background&color). As the reviewer suggested, we are working on an evaluation of our algorithm in DMControl (requiring less than 1M steps to master tasks). According to our primitive investigation in DMControl, our main findings in ProcGen (including the existence of interference) seem to remain the same. Indeed, in Figure 2b of [20; Laskin&Lee et al. NeurIPS 2020], color-related augmentations (random-conv, color-jitter) disturb RL training and lead to a less effective attention map (perhaps, poorer performance) than one with no augmentation. This suggests our techniques to reduce the interference (InDA or ExDA).
>
> **Weakness2**: Typo correction \
> We thank the reviewer tBmt for pointing out the typo in L189. $N_k(s)$ is the number of *times* selecting $\phi_k$ up to round $s$.
>
> **Question1**: Further discussion on DrAC+PAGrad \
> DrAC+PAGrad is dedicated to mitigating the interference between RL and DA losses ($L_{\text{PPO}}$ and $L_{\text{DA}})$ when optimizing them *concurrently*. This is an interesting approach itself since it is a new approach to reduce interference (compared to DrAC) in Figure 3. However, it concurrently optimizes the two losses and thus generates a substantial interference compared to the proposed methods (InDA or ExDA) that *explicitly* separates the two losses. In the revision, we will include the above discussion more clearly.
>
> **Question2**: The rationale for removing the inconsistency measure of value function from the distillation loss $L_{\text{DA}}$ \
> We exclude the value inconsistency loss from $L_{\text{DA}}$ since the value consistency is not necessary for RL training or distillation in the actor-critic framework and including it has a potential risk of generating additional interference. Indeed, we empirically observed that including the value inconsistency not only interferes more but also requires extra efforts to find a hyperparameter stabilizing training. We will clarify this in the revision.

---

> > ### Comment · Reviewer_tBmt · 2022-08-08
> > **response to author rebuttal**
> >
> > - Nice, it's good to see if the findings from ProcGen translate to DMC-- do the environments provide similar challenges that we are interested in as in the case of ProcGen? In the future, this could enable faster iterations.
> > - Thanks for the clarification on DrAC+PAGrad.
> >
> > It would be great if the clarifications in the revision are highlighted using a different colour so it is easier to understand what changes have been made.

---

> > > ### Author Response · Authors · 2022-08-09
> > > **Response to reviwer tBmt**
> > >
> > > Thank you for your advice, we highlight revision part using blue color.
> > >
> > > We experiment on DMC about ablation study of ExDA, but, unfortunately, we cannot do all the experiments shown in Procgen due to the limited time and computation resources, only a preliminary experiment providing a justification of our proposed method on DMC has been added to the revision (Appendix L). We first note that DrAC paper has demonstrated that UCB-DrAC (and thus UCB-InDA) can accelerate RL training in DMC. Hence, in order to justify our proposed method (UCB-ExDA = UCB-InDA + ExDA $\approx$ UCB-DrAC + ExDA), it would be sufficient to show the existence of the cases showing a benefit of ExDA in DMC. Hence, we have conducted an additional experiment on DMC.

---

### Author Response · Authors · 2022-08-09
**Interim revision uploaded**

We thank all the reviewers for the thorough reviews and the precious advices. We have been revising our manuscript accordingly. Due to the limited time and computation resources, we uploaded an interim version. However, the current one includes preliminary experiments, which we additionally conducted during the rebuttal period, justifying our proposed method (in particular, DA and ExDA) in DeepMind Control Suite with Soft Actor-Critic (i.e., other benchmark and algorithm), and demonstrating the robustness of the proposed method against a wrong augmentation. The updated part is shown in the color blue.

We would like to thank all the reviewers again. We will keep improving it to include all rebuttal discussions and to improve the presentation accordingly. We will sincerely appreciate any further suggestions or comments.

---

### Meta-Review · Area_Chair_qyUG · 2022-08-31

**Recommendation:** Accept
**Confidence:** Certain

**Metareview:**

Although data-augmentation is  now a common practice in Deep RL  this submission provides a new RL-friendly approach to how and when use augmentation in Deep RL, supported by extensive experimental study, which is a non-trivial contribution to the field.

**Award:**

No

---

### Decision · Program_Chairs · 2022-09-14

Accept